# Enzyme-controlled, nutritive hydrogel for mesenchymal stromal cell survival and paracrine functions

Cyprien Denoeud[1,5], Guotian Luo [1,5], Joseph Paquet[1], Julie Boisselier[2], Pauline Wosinski[1], Adrien Moya[1], Ahmad Diallo[1], Nathanael Larochette[1], Stéphane Marinesco [3], Anne Meiller [3], Pierre Becquart[1], Hilel Moussi[1], Jean-Thomas Vilquin[4], Delphine Logeart-Avramoglou [1], Adeline Gand[2], Véronique Larreta-Garde[2], Emmanuel Pauthe[2], Esther Potier[1] & Hervé Petite [1✉]

Culture-adapted human mesenchymal stromal cells (hMSCs) are appealing candidates for regenerative medicine applications. However, these cells implanted in lesions as single cells or tissue constructs encounter an ischemic microenvironment responsible for their massive death post-transplantation, a major roadblock to successful clinical therapies. We hereby propose a paradigm shift for enhancing hMSC survival by designing, developing, and testing an enzyme-controlled, nutritive hydrogel with an inbuilt glucose delivery system for the first time. This hydrogel, composed of fibrin, starch (a polymer of glucose), and amyloglucosidase (AMG, an enzyme that hydrolyze glucose from starch), provides physiological glucose levels to fuel hMSCs via glycolysis. hMSCs loaded in these hydrogels and exposed to near anoxia (0.1% pO$_2$) in vitro exhibited improved cell viability and angioinductive functions for up to 14 days. Most importantly, these nutritive hydrogels promoted hMSC viability and paracrine functions when implanted ectopically. Our findings suggest that local glucose delivery via the proposed nutritive hydrogel can be an efficient approach to improve hMSC-based therapeutic efficacy.

[1] University Paris Cité, CNRS, INSERM, ENVA, B3OA, Paris, France. [2] Biomaterial for Health Group, ERRMECe, University of Cergy-Pontoise, Cergy-Pontoise, France. [3] Neuroscience Research Center, AniRA-NeuroChem Platform, Lyon, France. [4] Sorbonne Université, INSERM, AIM, CNRS, Centre de Recherche en Myologie, Hôpital Pitié Salpêtrière, Paris, France. [5] These authors contributed equally: Cyprien Denoeud, Guotian Luo. ✉email: herve.petite@univ-paris-diderot.fr

There is tremendous excitement regarding the use of culture-adapted mesenchymal stromal cells (MSCs) for regenerative medicine based on their ability to proliferate, differentiate, and secrete a range of growth, angiogenic, and immune regulating factors. However, it is essential to temper hype with reality as most therapies' initial "proof of concept" has not yet been translated into routine clinical practices[1–3]. Specifically, MSC-based therapies delivered either as avascular tissue constructs generated in vitro or single cells directly injected into a wound area encounter a significant roadblock: the occurrence of massive MSC death post-implantation[4–6]. The lack of functional vascularization in tissue constructs or wound area, indeed, exposes implanted MSCs to the rigors of an ischemic milieu, a prime cause for the observed massive cell death post-implantation[7].

Scaffolds for delivering MSCs have been optimized to allow cell attachment, to deliver bioactive chemical compounds, and to expose MSCs to specific mechanical and biological stimuli[8,9]. Surprisingly, they rely, at best, on optimized scaffold diffusion characteristics (governed by scaffold porosity and permeability[10]) to enable diffusion of oxygen and nutrients and, thus, provide vital energy to the cells[11]. Furthermore, pre-vascularized engineering hydrogel has been produced and investigated for the improvement of oxygen and nutrient delivery[12–14]. The energy delivery strategy is far from optimal considering that MSCs are often transplanted into ischemic territories in large numbers where they must struggle for energy in a crowded and nutrient-poor tissue environment. The energy issue is exacerbated further by the low energy autonomy of MSCs[15]. hMSCs loaded in fibrin hydrogels and implanted subcutaneously in mice, indeed, showed a 66% and 40% decrease in glycolytic and ATP reserves, respectively, within 24 h and died massively (–94%) during the first 3 days of implantation[15]. These considerations call for engineering scaffolds that provide the necessary nutrients to MSCs for prolonging their viability post-implantation.

To compensate for hMSC low energy reserves and ultimately enhance clinical potency, we hereby propose to engineer a scaffold integrated with an inbuilt nutrient supply system. Our hypothesis is that matching MSC nutritive requirements would favor their survival and enhance their paracrine functions pertinent to angiogenesis. A rational engineering of such scaffolds requires a thorough understanding of their specific metabolic needs. To this aim, our research team[15–17] and others[18] demonstrated that hMSCs exposed to severe, continuous near-anoxia, but not glucose shortage, remained viable and maintained both their in vitro proliferative ability and, most importantly, their functions pertinent to tissue repair in vivo. These findings challenged traditional views according to which a lack of oxygen per se is responsible for the massive death of MSCs observed upon implantation and provided evidence that these cells can withstand exposure to near-anoxia provided that a glucose supply is available. These observations justify engineering scaffolds with an inbuilt glucose delivery system.

Storing and delivering glucose to the implanted MSCs is a challenge because high concentrations of this water-soluble molecule can disturb the osmotic pressure, causing cell shrinkage[19]. To resolve this matter, we took inspiration from nature and developed an enzyme-controlled glucose-delivery fibrin hydrogel. In this system, large amounts of glucose are stocked within the fibrin hydrogel in a compact and osmotically inert manner by using starch, the energy storage polysaccharide found in plants[20] The delivery of glucose is achieved by the enzymatic hydrolysis of starch by amyloglucosidase (AMG) (Fig. 1a). The present study demonstrates for the first time that starch/AMG hydrogels can provide glucose for in situ "fueling" MSCs and, thus, improving their survival and paracrine functions both in vitro under near anoxia and in vivo post-implantation.

## Results

**The starch/AMG hydrogels produce glucose**. We first determined glucose production from starch/AMG hydrogels (Fig. 1b) using a glucose microelectrode biosensor placed in the center of hydrogel. In the presence of AMG, the starch hydrogels produced glucose as a function of starch concentration (Fig. 1b). Glucose production plateaued within 400 to 600 s and was maximal with 2% starch concentration ($3.76 \pm 0.31$ mM; Fig. 1b). In contrast, the absence of AMG led to no detectable glucose production (Fig. 1b). Moreover, glucose production inside 2% starch/AMG hydrogels was linearly proportional to AMG concentration ($R^2 = 0.977$ at 180 s; Fig. 1c). Environmental scanning electron microscopy (eSEM) studies revealed that glucose-free hydrogels formed a fibrillar network. "Open-cell" structures and interconnected pores of irregular shapes formed when glucose, starch or starch/AMG were added, with starch addition leading to a higher pore size (Fig. 1d).

**The starch/AMG hydrogels improve the survival of hMSCs in near anoxia in vitro**. We next determined whether the starch/AMG hydrogels sustained the viability of hMSCs. To this aim, hydrogels with increasing starch concentrations either with (Fig. 2a) or without (Fig. 2b) AMG were loaded with $10^5$ hMSCs and maintained in glucose-free and serum-free media under near anoxia (0.1% $pO_2$). On day 7 and 14, all samples were harvested and digested to determine the number of viable hMSCs using flow cytometry following staining with Hoechst 33342 and propidium iodide. On day 7, the number of viable cells inside the starch/AMG hydrogels was linearly proportional to the initial starch concentration ($R^2 = 0.984$) (Supplementary Fig. 1a). On day 14, the number of hMSCs was still high inside the 2% starch/AMG hydrogels, reaching 66%. In contrast, the number of viable cells in the starch hydrogels without AMG was drastically reduced at day 7, with almost no viable hMSCs remaining at day 14 (Fig. 2a versus 2b). These results provided evidence of the critical role of AMG in achieving the glucose concentrations required for ensuring the long-term survival of hMSCs. The 2% starch/AMG hydrogel was chosen for the rest of the present study because it was the most effective hydrogel for enhancing hMSC survival. To further investigate the functionality of the hMSCs in the 2% starch/AMG hydrogels after 14 days of culture in near anoxia, their proliferative ability was compared to that obtained from naive hMSCs cultured in standard conditions over 10 days. The cell doubling time of these two populations was similar ($p > 0.05$) (Supplementary Fig. 1b). Moreover, these two cell populations were indistinguishable in terms of CD marker expression: specifically, both were positive for CD73, CD90, CD105, and negative for CD45 (Supplementary Fig. 1c). Taken together, these data provided evidence that the starch/AMG hydrogels enhanced hMSC survival and maintain their functionality (mirrored by their proliferative ability and CD marker expression) in near anoxia.

**The 2% starch/AMG hydrogels are more effective than glucose hydrogels in extending the survival of hMSCs in near anoxia in vitro**. We assessed the efficacy of 2% starch/AMG hydrogels in extending hMSC survival compared to control hydrogels. On day 7, the number of viable hMSCs loaded in the 2% starch/AMG hydrogels was 4-fold higher compared to the cells loaded in 5.5 mM glucose hydrogels. On day 14, viable hMSCs were only observed in the 2% starch/AMG hydrogels (Fig. 2c).

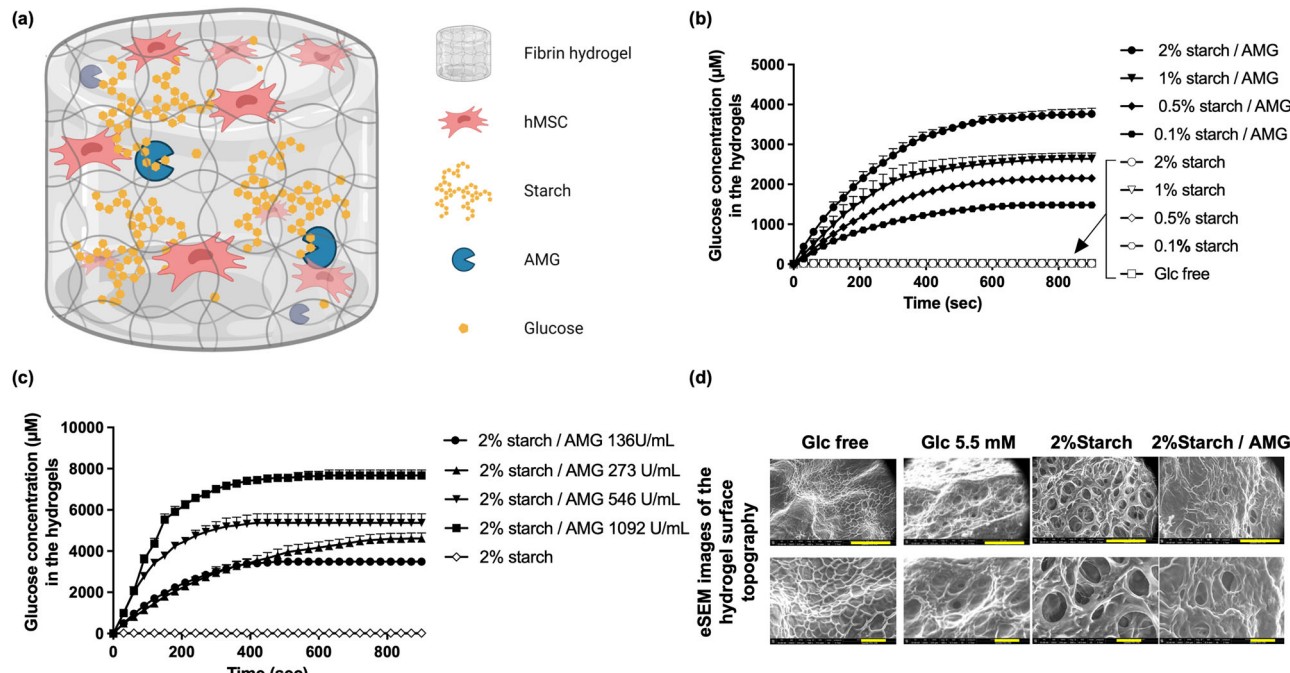

**Fig. 1 The starch/AMG hydrogels produce glucose. a** Schematic (not to scale) of the starch/AMG hydrogels. hMSCs were embedded in fibrin hydrogels containing starch enzymatically hydrolyzed by AMG in glucose, fueling hMSCs. (Created with BioRender.com) **b**. Production of glucose in hydrogels with increasing concentrations of starch loaded with or without AMG ($n = 5$–7). **c** Production of glucose in 2% starch hydrogels loaded with increasing concentrations of AMG ($n = 4$–8). **d** Representative environmental scanning electron micrographs (eSEM) micrographs of the external architecture of either glucose-free, 5.5 mM glucose, or starch (with or without AMG) hydrogels. Scale bars: 200 µm (for eSEM images in the top row), 50 µm (for eSEM images in bottom row). AMG amyloglucosidase, Glc glucose.

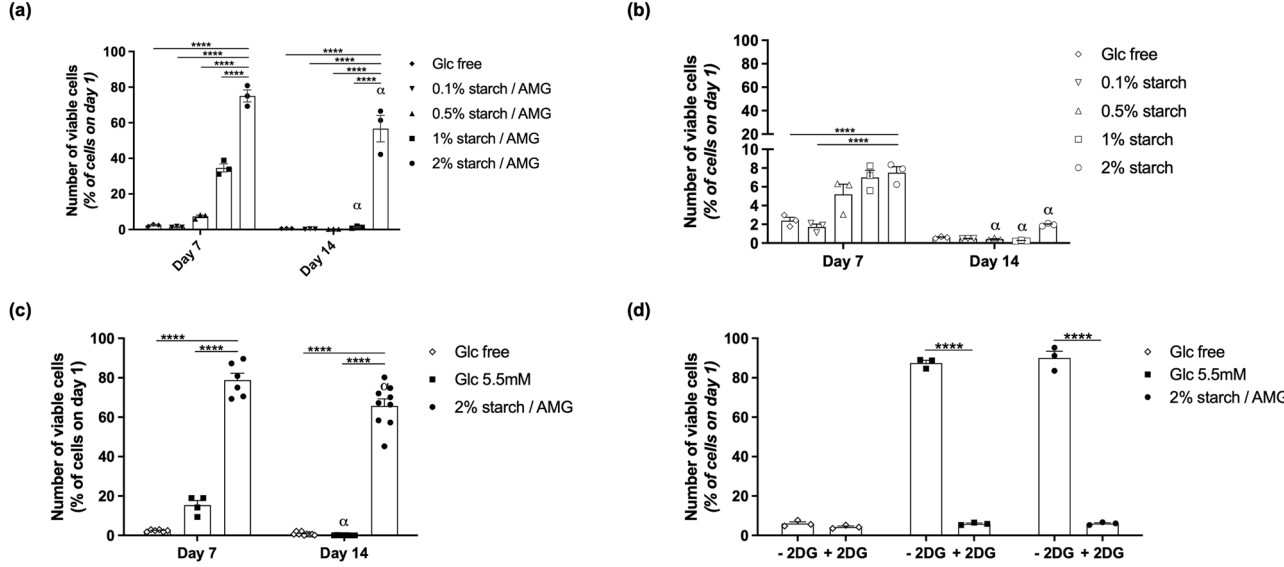

**Fig. 2 The starch/AMG hydrogel environment extends survival of hMSCs by delivering glucose in near anoxia in vitro. a**, **b** Viability of hMSCs seeded in either glucose-free hydrogels or hydrogels with increasing starch concentrations either with **a** or without **b** AMG, after exposure to near-anoxia (0.1% pO$_2$). ($n = 3$). **c** Viability of hMSCs seeded in either glucose-free, 5.5 mM glucose, or starch 2%/AMG hydrogels after exposure to near-anoxia for 7 and 14 days. ($n = 4$–9). **d** Viability of hMSCs seeded in either glucose-free, 5.5 mM glucose, or starch 2%/AMG hydrogels after exposure to 2-Deoxy-D-Glucose (a competitive inhibitor for the production of glucose-6-phosphate) in near-anoxia for 3 days. ($n = 3$).

To determine whether the glucose released from the hydrogels was responsible for the survival of hMSCs under near anoxia via glycolysis, 2-Deoxy-D-Glucose (a competitive inhibitor for the production of glucose-6-phosphate from glucose at the phosphoglu-coisomerase level) was added to the 2% starch/AMG, glucose-free, and 5.5 mM glucose hydrogels loaded with 10$^5$ hMSCs. Massive cell death was observed in all the conditions tested within 3 days of exposure to 2-Deoxy-D-Glucose (Fig. 2d) providing evidence that the glucose released from 2% starch/AMG or 5.5 mM glucose hydrogels was responsible for ensuring hMSC survival via glycolysis.

Overall, these results demonstrate that glucose was central to hMSC survival maintenance, and that, compared to results

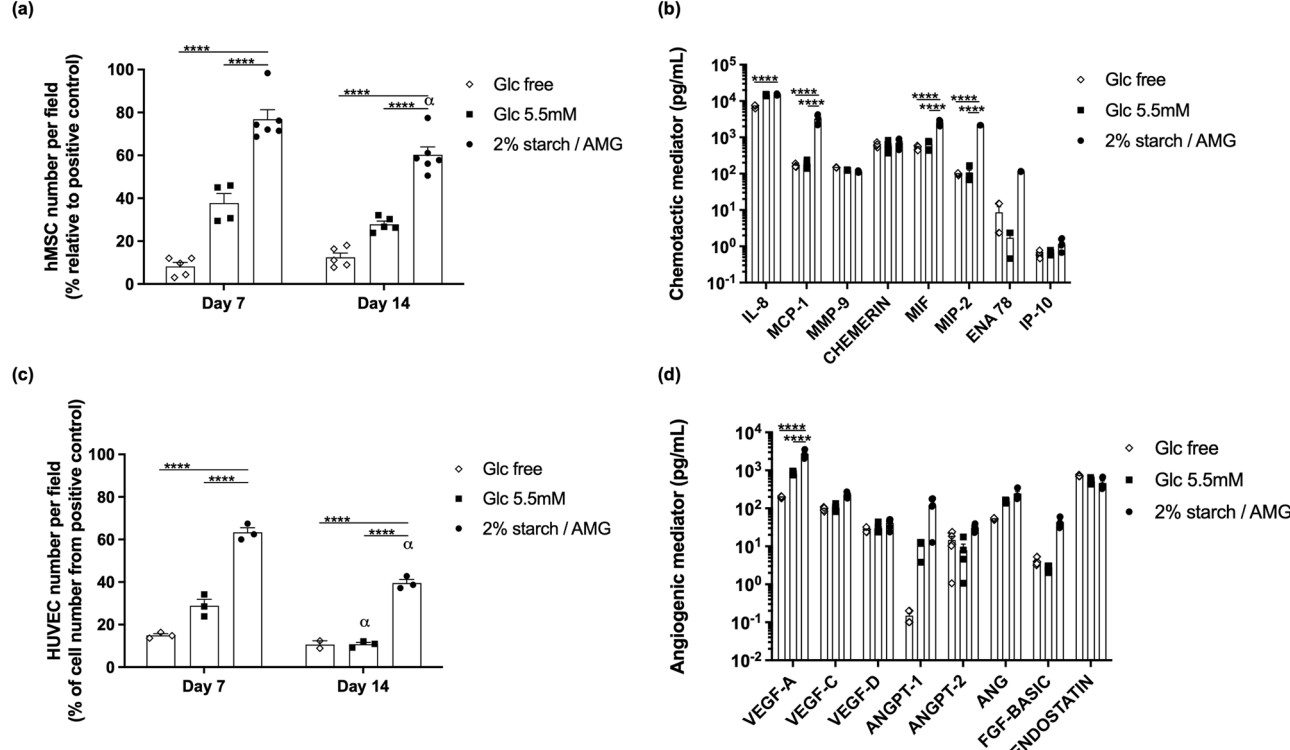

**Fig. 3 The starch/AMG hydrogels are more effective than glucose hydrogels in enhancing the viability and chemotactic functions of hMSCs in near anoxia in vitro. a** Quantification of hMSC migration in Boyden chambers in response to conditioned media from hMSCs seeded in either glucose-free hydrogels, 5.5 mM glucose hydrogels, or starch 2%/AMG hydrogels and exposed to near-anoxia for 7 and 14 days. ($n = 4$–6) **b**. Concentrations of the chemotactic mediators in the conditioned media from hMSCs seeded in either glucose-free, 5.5 mM glucose, or starch 2%/AMG hydrogels and exposed to near-anoxia for 7 and 14 days. ($n = 3$). **c** Quantification of HUVEC migration in Boyden chambers in response to conditioned media from hMSCs seeded in either glucose-free, 5.5 mM glucose, or starch 2%/AMG hydrogels, after exposure to near-anoxia conditions for 7 and 14 days. ($n = 3$) **d**. Concentrations of the proangiogenic mediators in the conditioned media from hMSCs seeded in either glucose-free, 5.5 mM glucose, or starch 2%/AMG hydrogels and exposed to near-anoxia for 7 and 14 days. ($n = 3$). IL interleukin, MCP monocyte chemoattractant protein, MMP matrix metalloproteinase, MIF macrophage inhibitory factor, MIP macrophage inflammatory protein, ENA epithelial neutrophil activating protein, IP interferon gamma-induced protein, VEGF vascular endothelial growth factor, ANGPT angiopoietin, ANG angiogenin, FGF-BASIC basic fibroblast growth factor.

obtained with the 5.5 mM glucose hydrogels, the survival of hMSCs in near anoxia was improved when the cells were seeded within the 2% starch/AMG hydrogels.

**The starch/AMG hydrogels are more effective than glucose hydrogels in enhancing the chemotactic and proangiogenic functions of hMSCs in near anoxia in vitro**. The chemo-attraction of progenitors (such as MSCs) and endothelial cells is relevant to the success of MSC-based regenerative medicine. We, therefore, compared the chemotactic potential of supernatant conditioned medium (CM) collected from hMSCs seeded into either glucose-free, 5.5 mM glucose or 2% starch/AMG hydrogels towards hMSCs or human umbilical vein endothelial cells (HUVECs) using the Boyden chamber migration assay and cultured for 7 and 14 days in near anoxia.

Compared to CM from hMSCs seeded in either glucose-free or in 5.5 mM glucose hydrogels, CM from hMSCs seeded in 2% starch/AMG hydrogels promoted a significant ($p < 0.0001$) increase in the chemotactic potential towards hMSCs (9.3- and 2.0-fold increase, respectively, after 7 days; 4.8- and 2.7- fold increase, respectively, after 14 days; Fig. 3a). Moreover, CM from hMSCs seeded in 2% starch/AMG hydrogels exhibited a significant increase in MCP-1, MIF and MIP-2 when compared to 5.5 mM glucose hydrogels ($p < 0.0001$) and a significant increase in Il-8, MCP-1, MIF, MIP-2 when compared to glucose-free hydrogels ($p < 0.0001$) after 14 days of culture(Fig. 3b).

Compared to CM from hMSCs seeded in either glucose-free or 5.5 mM glucose hydrogels, the CM from hMSCs seeded in 2% starch/AMG hydrogels promoted a significant ($p < 0.0001$) increase in the chemotactic potential towards HUVECs (4.2- and 2.2-fold increase, respectively after 7 days; 3.7- and 3.7-fold increase, respectively, after 14 days; Fig. 3c). Moreover, the concentration of the proangiogenic mediators VEGF-A was significantly increased in the CM from hMSCs seeded in 2% starch/AMG hydrogels compared to CM from glucose-free and 5.5 mM glucose hydrogels after 14 days of culture (Fig. 3d) ($p < 0.0001$). We also found increased amount of VEGF-C, VEGF-D, ANGPT-1, ANGPT-2, ANG, and FGF-basic, although not by a statistically significant quantity in 2% starch/AMG hydrogels when compared to other hydrogels (Fig. 3d). Interestingly, endostatin, an angiogenesis inhibitor, exhibited a 1.5 trend ($p < 0.05$) towards an increase in CM from hMSCs seeded in glucose-free hydrogels compared to CM from 2% starch/AMG hydrogels after 14 days of culture (Fig. 3d). Altogether, these data provided evidence that the 2% starch/AMG hydrogels promoted the in vitro chemotactic and proangiogenic functions of hMSCs cultured in near anoxia.

**The starch/AMG hydrogels are more effective than glucose hydrogels to increase the in vivo survival of hMSCs after subcutaneous implantation**. To assess first the glucose production in vivo, acellular hydrogels were implanted ectopically in

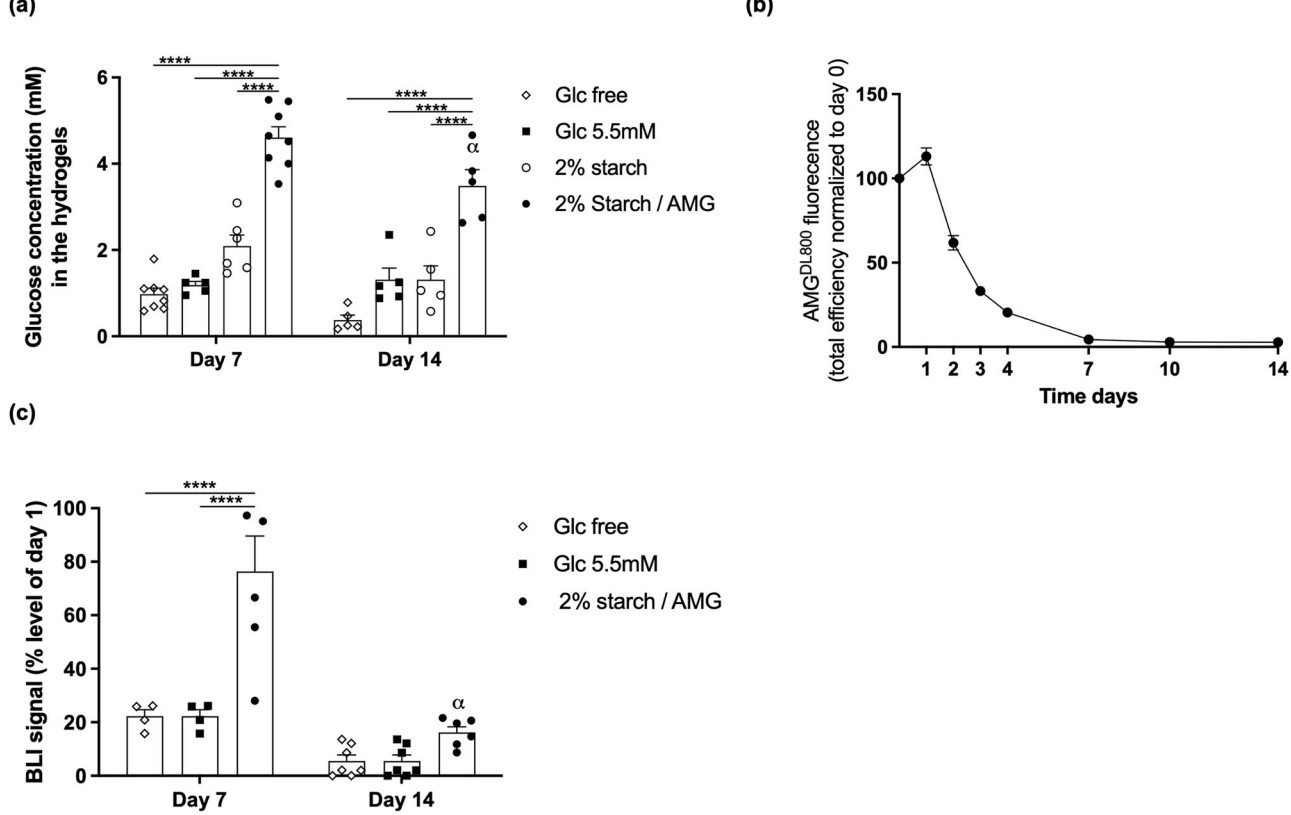

**Fig. 4 The starch/AMG hydrogels are more effective than glucose hydrogels in increasing the in vivo survival of hMSCs after subcutaneous implantation. a** Glucose concentration into glucose-free, 5.5 mM glucose, or 2% starch cell free hydrogels with or without AMG after subcutaneous implantation in nude mice for 7 and 14 days ($n = 5$–8). **b** Kinetics of AMG fluorescence efficiency after subcutaneous implantation in nude mice up to 14 days ($n = 6$). **c** Quantification of the survival of Luc-ZSGreen-hMSCs loaded into glucose-free, 5.5 mM glucose, or starch 2%/AMG hydrogels and implanted in an ectopic mouse model through bioluminescent signal monitoring ($n = 4$-7).

mouse and intra-hydrogel glucose concentration was determined on day 7 and 14. At both time points, the intra-hydrogel glucose concentration was significantly ($p < 0.0001$) higher inside the 2% starch/AMG hydrogels compared to results obtained with the 2% starch hydrogels (4.6- and 9.4-fold-increase, respectively), glucose-free hydrogels (3.9- and 2.9-fold-increase, respectively), and 5.5 mM glucose hydrogels (2.3- and 2.7-fold-increase, respectively) (Fig. 4a). Moreover, a significant decrease in intra-hydrogel glucose concentration inside the 2% starch/AMG hydrogels was observed between day 7 and 14 suggesting a cessation of glucose production. The termination of AMG-mediated glucose production was corroborated by monitoring the fate of fluorescence-tagged AMG loaded into acellular hydrogels which showed evidence of its quasi-disappearance at day 7 of implantation (Fig. 4b).

We then assessed the ability of the different hydrogels to enhance the survival of Luc-ZSGreen-hMSCs in an ectopic transplantation mouse model. After 7 days, Luc-ZSGreen-hMSCs seeded within the 2% starch/AMG hydrogels exhibited a significant ($p < 0.0001$) increase in BLI signal compared to the signals obtained with either glucose-free or 5.5 mM glucose hydrogels (3.4- and 2.8-fold increase, respectively). At day 14, Luc-ZSGreen-hMSCs seeded within the 2% starch/AMG hydrogels exhibited a significant decline in BLI signal compared to day 7, underscoring the critical role of AMG in timely delivery of glucose to transplanted hMSCs (Fig. 4c). These observations were further validated by counting the numbers of hβ2-MG-positive cells on histological sections (Supplementary Fig. 1d).

Taken together, these data provided evidence that the 2% starch/AMG hydrogels enhanced hMSC survival after ectopic implantation in mice for up to 7 days.

**The starch/AMG hydrogels are more effective than glucose hydrogels in increasing the proangiogenic functions of hMSCs after subcutaneous implantation**. hMSC proangiogenic functions were investigated by loading hMSCs into either glucose-free, 5.5 mM glucose, or 2% starch/AMG hydrogels. Each hydrogel was first placed in the middle of a silicone cylinder and acellular fibrin hydrogel was casted on both sides (Fig. 5a). This "sandwich" construction was then implanted in an ectopic mice model. The volume occupied by newly-formed blood vessels in the acellular fibrin portions was quantified after 14 and 21 days of implantation. The new blood vessels formed in the vicinity of the 2% starch/AMG hydrogels filled a 2.9- and 4.0-fold increased volume compared to results obtained from glucose-free hydrogels at day 14 ($p < 0.01$) and 21 ($p < 0.0001$), and a 3.4- and 2.3-fold increased volume compared to results obtained from 5.5 mM glucose hydrogels at day 14 ($p < 0.01$) and 21 ($p < 0.0001$), respectively (Fig. 5b, c). Similarly, the new blood vessels formed near the 2% starch/AMG hydrogels were significantly longer than those found near the glucose-free hydrogels on day 14 ($p < 0.05$) and 21 ($p < 0.01$) and the 5.5 mM glucose hydrogels on day 21 ($p < 0.01$), respectively (Fig. 5b, d). In contrast, the new blood vessel diameter and blood vessel number were similar for all tested hydrogels (Fig. 5d).

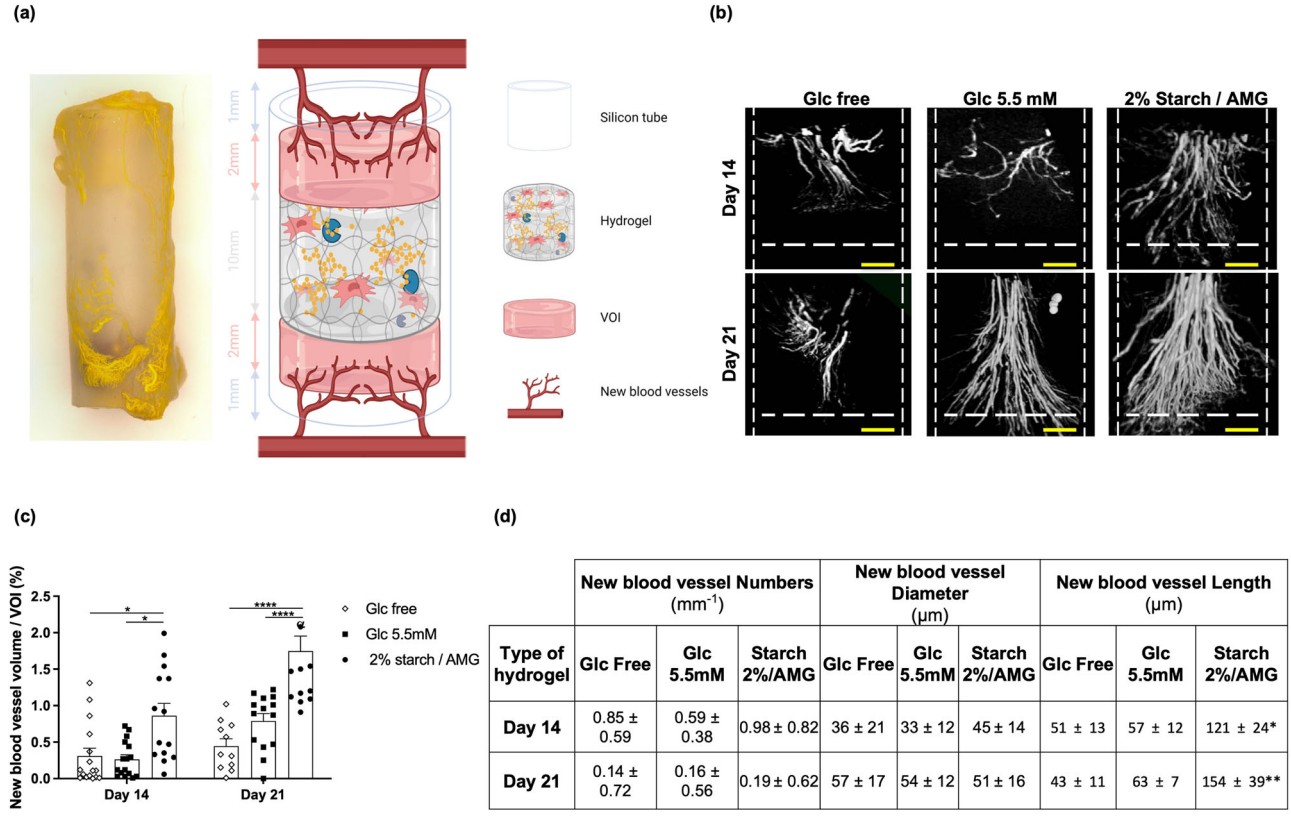

**Fig. 5 The starch/AMG hydrogels are more effective than glucose hydrogels in increasing the in vivo proangiogenic functions of hMSCs after subcutaneous implantation. a** A gross view of the hydrogel and schematic (not to scale) of the experimental model designed to assess the proangiogenic potential of hMSC-containing hydrogels. (Created with BioRender.com). **b, c** Micro-CT vasculature analysis in the vicinity of either glucose-free, 5.5 mM glucose, or starch 2%/AMG hydrogels after 14 and 21 days of ectopic implantation in mice. **b** Representative 3D reconstructions of the new blood vessels into the volume of interest (VOI) of either glucose-free, 5.5 mM glucose, or starch 2%/AMG hydrogels after subcutaneous implantation in nude mice, scale bars: 500 µm. **c** Quantification of new blood vessel volume into the VOIs of either glucose-free, 5.5 mM glucose, or starch 2%/AMG hydrogels after subcutaneous implantation in nude mice ($n = 8$). **d** Table summarizing the number, diameter, and length of newly formed blood vessels into the VOIs of either glucose-free, 5.5 mM glucose, or starch 2%/AMG hydrogels after subcutaneous implantation in nude mice ($n = 8$).

Overall, these data demonstrated that the 2% starch/AMG hydrogels promoted the proangiogenic function of hMSCs for up to 21 days after subcutaneous implantation in mice.

## Discussion

Upon implantation, survival and maintenance of functional hMSCs are the most critical challenges to meet for engineering successful tissue constructs of clinically relevant volume. The use of growth factors or cells to stimulate angiogenesis post-implantation and prevascularized tissue constructs has been proposed to overcome these major roadblocks[21,22]. Their successful translation in routine clinical practice is, however, still pending. To enable hMSCs to survive and contribute through their paracrine effects to the regenerative process, we hereby propose a new approach based on the fulfillment of their metabolic needs. To do so, we developed a starch/AMG hydrogel that enabled the supply of glucose levels sufficient to ensure their survival and paracrine functions at least over 7 days after implantation.

The present report provides evidence that starch/AMG hydrogels release glucose as a function of starch and AMG concentrations (Fig. 1b, c). In addition, glucose production plateaued within 400 to 600 s, reflecting the balance between new glucose production and glucose diffusion out of the hydrogel. Interestingly, cell-free starch/AMG hydrogels explanted on days 7 and 14 exhibited glucose concentrations known to support hMSC proliferative capacity or ability to produce growth factors (Fig. 4a)[23].

Starch/AMG hydrogels significantly ($p < 0.0001$) extended the survival of hMSCs in vitro under near anoxia compared to glucose-free and 5.5 mM glucose hydrogels (Fig. 2a). Most importantly, the maintenance of hMSC proliferative ability and CD marker expression after their release from these starch/AMG hydrogels bodes well for their continued in vivo therapeutic efficacy (Supplementary Fig. 1b, c). A demonstration of the maintenance of the multipotential and immunosuppressive character of MSCs under these conditions will enable these results to be extended to all the classically recognized functionalities of MSCs. Another significant result was that loading either adipose-derived hMSCs (ADSCs) or human myoblasts in starch/AMG hydrogels resulted in a 200- and 100-fold increase of cell survival compared to results from glucose-free hydrogels after 14 days in near anoxia (Supplementary Fig. 2a, b), extending the potential applications of the proposed strategy to additional progenitor cells widely used in regenerative medicine. In this near anoxia context, glycolysis was the main energy-provider pathway to hMSCs as blocking this pathway resulted in massive cell death (Fig. 2d), corroborating previous results establishing that hMSCs exclusively rely on the glycolytic pathways under near anoxia[15,17]. Most importantly, after implantation in an ectopic mouse model, hMSCs embedded into starch/AMG hydrogels exhibited a 3.5- and a 2.8-increase in hMSC viability compared to glucose-free and 5.5 mM glucose hydrogels at day 7 (Fig. 4c). These data were further confirmed by counting hMSCs on histological sections (Supplementary Fig. 1d). However, hMSC viability declined

between day 7 and 14 of implantation. This decline in the performance of starch/AMG hydrogels could be attributed to starch stock exhaustion. However, the presence of starch remnants at day 14 post-implantation (Supplementary Fig. 3) contradicts this hypothesis. This starch/AMG hydrogel performance loss may be attributed to the gradual disappearance of AMG (Fig. 4b), which, in turn, leads to a drop in glucose production, an unfulfillment of MSCs energy requirements and, ultimately to hMSC viability loss. Further research is needed to determine whether this gradual disappearance of free AMG is due to its diffusion from the hydrogel or to its degradation by local proteases. These observations also prompt the development of immobilization strategies to confine AMG in hydrogels and improve its stability[24]. Another tenable explanation for the observed decrease in hMSC viability over time is that the implanted engineered constructs, which temporarily disrupt homeostasis, induce a local exacerbated immune response whereby the host immune system recognizes hMSCs as foreign, ultimately leading to hMSC death. Although further studies are needed to exclude this hypothesis, the immunocompromised features of the mice, the parallel disappearance of AMG and MSCs and the key role of glucose in survival, chemotactic and proangiogenic functions of hMSCs make it unlikely. Alternatively, the foreign body reaction may lead to chronic fibrotic capsule formation[25,26], preventing hydrogel revascularisation. Again, the improved chemotactic and proangiogenic functions of hMSCs do not support such a scenario.

Dead or dying MSCs, however, have been shown to be therapeutically beneficial[27]. In line with these studies, CM from cellularized glucose-free scaffolds displayed a biologically relevant level of chemotactic and angiogenic mediators (Fig. 3b, d) that translated into a chemotactic potential towards hMSCs and HUVECs (Fig. 3a, c). Cellularized glucose-free scaffolds also showed some new blood vessel formation in vivo, but whether they perform better than acellular glucose-free scaffolds cannot be inferred from the present study. Most importantly, and in agreement with previous studies demonstrating that viable MSCs performed better than dead cells[17,28–30], we hereby observed a significant viability-dependent effect on most chemotactic (MCP-1, MIF, MIP-2, and Il-8) and proangiogenic (VEGF-A) mediator concentrations (Fig. 3b, d). The biological relevance of these results was supported by the observed enhancement of the CM chemotactic potential towards hMSCs and HUVECs (Fig. 3a, c) in vitro. It was further substantiated by our in vivo findings which revealed a significant increase in the volume of new blood vessel induced by the starch/AMG hydrogels loaded with hMSCs when compared to control hydrogels (Fig. 5b, c). Interestingly, this increase in vessel volume results in an increase in vessel length rather than an enlargement in vessel diameter.

The delivery of glucose rather than oxygen offers a paradigm shift in the strategy for combating MSC death post-implantation. Unlike oxygen delivery systems that may delude MSCs by mimicking a physioxic environment and may compromise their ability to mount a response to hypoxia, the proposed starch/AMG technology aims to provide MSCs with sufficient glucose while maintaining exposure to hypoxia, an essential catalyst for angiogenic factors release to accelerate revascularisation. However, this strategy is still in its infancy and to reach their full potential, hydrogels will have to be fully resorbable and tailored to the dose of MSCs and the duration of treatment required, as with any pharmacological treatment.

In conclusion, relying exclusively on the capacity of the recipient bed to supply glucose at a level that meets the MSC needs is, in all likelihood, not sufficient when a large number of cells is administered at one time. Prevailing approaches proposed to prevascularize tissue constructs. We hereby provide, for the first time, a novel alternative based on the delivery of glucose to transplanted cells. In this system, the delivery of glucose was achieved from a starch polymer by enzymatic hydrolysis (AMG). Released glucose-fueled hMSCs and improved their survival and paracrine functions post-implantation. As glucose is the main source of energy to power most mammalian cells, these new glucose-delivering hydrogels should ultimately have much wider applications in regenerative medicine.

## Methods
All chemicals were purchased from Sigma-Aldrich and used as received unless otherwise specified.

**Culture of human mesenchymal stem cells (hMSCs).** HMSCs were isolated from bone marrow obtained as discarded tissue during routine surgery from four adult donors at the Lariboisiere Hospital, Paris, France. The tissues of interest were collected with the patient's informed consent and agreed with Lariboisiere Hospital regulations. All ethical regulations relevant to human research participants were followed. HMSCs were isolated from each donor's bone marrow using a procedure adapted from literature reports[31,32], characterized by CD marker expression and differentiation potential, and cultured in Alpha Minimum Essential Medium (α-MEM; PAN Biotech GmbH, Aidenbach, Germany) supplemented with 10% Fetal Bovine Serum (FBS; PAA Laboratories GmbH, Les Mureaux, France) and 1% antibiotic/antimycotic (atb/atm; v/v, PAA Laboratories GmbH, Cörbe, Germany) under standard cell culture condition, that is, a humidified, 37 °C, 5% $CO_2$, and 95% air environment. When 80–90% confluence was reached, the hMSCs were trypsinized using trypsin-EDTA (Sigma-Aldrich, St Quentin Fallavier, France) and replated at a density of $10 \times 10^3$ cells/cm². HMSCs from the four donors were pooled at an equal ratio at passage 1 and further expanded to passage 4 before use.

**Transduction of hMSCs.** The pooled hMSCs (at passage 2) were transduced with a lentiviral vector encoding the firefly luciferase and the ZSGreen proteins (pRRLsin-MND-Luc-IRES2-ZS Green-WPRE; TRANSBIOMED, Bordeaux, France) as previously described[7] and further expanded. Flow cytometry analysis of ZSGreen-positive cells showed that 88% of the hMSCs were transduced. These cells will be referred to as "Luc-ZS Green-hMSCs" thereafter.

**Culture of Human Umbilical Vein Endothelial Cells (HUVECs).** HUVECs (Lonza, Walkersville, USA) were cultured in Endothelial Cell Growth Medium (EBM-2; Lonza, Levallois-Perret, France) supplemented with 5% FBS and 1% ATB/ATM, under standard cell culture conditions. When 80–90% confluence was reached, the HUVECs were trypsinized and replated at a density of $15 \times 10^3$ cells/cm². Cells at passage 4 were used for the in vitro experiments.

**Preparation of wheat starch solutions.** Wheat-derived starch solutions at either 0.2, 1, 2, or 4% (w/v) concentrations were prepared by stirring and heating starch powder in 10 mM HEPES buffer containing 0.3 M NaCl, 0.04 M $CaCl_2$ (pH 7.4) at 92 °C for 2 h.

**Preparation of hydrogels.** Hydrogels were prepared by mixing two aqueous solutions (Mix 1 and Mix 2) detailed in Supplementary Table 1. Whenever relevant, hMSCs were added in the mix 1 before polymerisation. The Mix 1 was first deposited on a hydrophobic surface (specifically, a polytetrafluoroethylene (PTFE), disk diameter = 6 mm), and the Mix 2 was then added

and homogenized within the Mix 1. These hydrogels were maintained at 37 °C in a humidified atmosphere for 1 h. After polymerization, each hydrogel was carefully removed from the PTFE surface, before use. Four types of cellularized hydrogels were prepared: (i) "Glc-free": glucose-free hydrogel that represents a negative/empty control hydrogel; (ii) "Glc 5.5 mM": glucose hydrogel loaded with 5.5 mM glucose that represents a standard control hydrogel; (iii) "Starch": fibrin hydrogels loaded with starch to reach the final starch concentration of 0.1%, 0.5%, 1% or 2%; and (vi) "Starch/AMG": fibrin hydrogels loaded with starch to reach the final starch concentration of 0.1%, 0.5%, 1% or 2% and amyloglucosidase (AMG). A concentration of 0.1% AMG was chosen because, of the AMG concentrations tested ranging from 136 U/ml to 1092 U/ml, a concentration of 136 U/ml AMG resulted in the slower degradation of starch in vitro. 100 µl hydrogels were loaded with $10^5$ hMSCs for the in vitro experiments. 200 µl hydrogels were loaded with $2 \times 10^5$ Luc-ZS Green-hMSCs for the in vivo assessment of hMSC survival. 125 µl hydrogels were loaded with $1.25 \times 10^5$ hMSCs for the in vivo assessment of the proangiogenic potential of the hMSC secretome. For the in vitro culture experiments, hydrogels were placed in individual wells of 12-wells tissue-culture plates, and immersed in 2 mL serum- and glucose-free α-MEM culture medium and placed in a humidified incubator at 0.1% oxygen tension using a proOx-C-chamber system (C-Chamber, C-374, Biospherix, New York, NY).

**Hydrogels surface topography.** The surface topography of the hydrogels was assessed using environmental scanning electron microscopy (Philips eSEM FEG/XL-30). The specimens were treated with liquid nitrogen for 30 s just before scanning to make the external hydrogel surface visible and then photographed.

**Hydrogel-contained glucose quantification without hMSCs.** The amount of glucose inside the hydrogels was quantified using a glucose electrode biosensor (AniRA-Neurochem platform, University Lyon I, France) placed at the center of the samples[33]. The glucose microelectrode biosensor was composed of a 25 µm diameter platinum wire insulated in a pulled glass capillary with a 100 µm long sensing tip coated with electropolymerized poly-m-phenylenediamine and with glucose oxidase enzyme immobilized with poly(ethylene glycol) diglycidyl ether (PEGDE)[33]. Glucose oxidation into gluconolactone by the enzyme produced hydrogen peroxide ($H_2O_2$), which was reoxidized at the platinum electrode, creating an oxidation current directly proportional to glucose concentration (based on standard curve).

**Near-anoxia culture conditions.** The "ischemic" environment was simulated in vitro by culturing the hydrogel-contained hMSCs under both near-anoxia (0.1% $O_2$) and nutrient (including both serum and glucose) deprivation[17]. Near-anoxia was achieved using a proOx-C-chamber system (C-Chamber, C-374, Biospherix, New-York, USA). The oxygen concentration in this chamber was maintained at 0.1%, with a residual gas mixture composed of 5% $CO_2$ and balanced nitrogen at 37 °C for the experiment duration. Environmental nutrient deprivation was achieved using a glucose-free α-MEM (PAN Biotech GmbH).

**Quantification of viable and apoptotic cells within hydrogels.** The viability of hMSCs maintained inside the hydrogels under near anoxia was evaluated on day 1, 7, and 14 after trypsinization using Hoechst 33342 (HE) and propidium iodide (PI) staining and attune flow cytometer (Life Technologies, Saint Aubin, France)[15]. In details, at the prescribed times, the hydrogels were incubated with both 1 µg/mL nucleic acid stain Hoechst 33342

(HE; Sigma-Aldrich) and 1 µg/mL propidium iodide (PI; Sigma-Aldrich) at 37 °C for 20 min; the hydrogels were then observed and photographed using a Zeiss confocal microscope (LSM 800, Zeiss, Göttingen, Germany). Hydrogels were then digested, and hMSCs were detached from hydrogels using trypsin-EDTA for 20 min. Then PBS containing 2% bovine serum albumin (BSA, Sigma-Aldrich) was added to stop the chemical action of trypsin. After centrifugation (at 3,500xg for 5 min), the hMSCs were resuspended in fresh PBS and analyzed using an Attune flow cytometer (Life Technologies, Saint Aubin, France). Cells staining both HE positive and PI negative were identified as "viable cells", whereas those staining both HE positive and PI-positive were identified as "dead cells". Cell viability was expressed as the number of viable cells at each point tested normalized with the respective viable cell number on day 1 of culture. For each hydrogel tested, cell viability on day 1 of culture was determined to be more than 90% of the $10^5$ hMSCs seeded at day 0.

**Proliferative potential of hydrogel-contained hMSC under near-anoxia.** After 14 days under near-anoxia, hMSCs were trypsinized from the 2% starch/AMG hydrogels and transferred back into tissue culture-flask ($10^3$ cells/cm²) and standard cell culture conditions (i.e., 21% $O_2$ and further cultured in α-MEM containing 5.5 mM glucose and 10% fetal bovine serum) for 10 days. The doubling time of these hydrogel-derived hMSCs was determined and compared to the one of naive hMSCs cultured under standard cell culture conditions for 14 days. The phenotype of these two hMSC population was also determined using flow cytometry as previously described[17].

**Preparation of hMSC conditioned media (CM).** The hydrogel-contained hMSCs were maintained under near anoxia with no cell culture medium change. On day 7 and 14, the supernatant cell culture media (further referred as conditioned media (CM)) were collected, centrifuged at $700 \times g$ for 5 min, aliquoted, and kept at −80 °C until further use.

**Assessment of the chemotactic effect of conditioned media.** The chemo-attractive potential of CM from hydrogel-contained hMSCs maintained under near anoxia for either 7 or 14 consecutive days was determined using the Boyden chamber migration assay as described previously[32]. The hMSCs or HUVECs that had migrated through the porous membrane were photographed using a Keyence VHX-2000F microscope (Courbevoie, France) and counted using *Image J* free software (National Institute of Health, Bethesda, USA). In details, 600 µL of each respective conditioned media (CM) were added to the bottom well of the Boyden chamber, and $2 \times 10^4$ hMSCs or $5 \times 10^4$ HUVECs were seeded on the top of the 8-µm-pore-diameter porous membrane (disk diameter = 6.5 mm; Transwell®; VWR International, Fontenay-sous-Bois, France), previously coated with 0.5% gelatin (Sigma-Aldrich) and cultured in glucose- and serum-free media. Serum-free α-MEM and α-MEM supplemented with 10% FBS were added to the bottom wells as negative and positive controls. After 6 h of maintenance under standard cell culture conditions, the cells on the top of the porous membrane were scrapped to remove the hMSCs or HUVECs that had not migrated from the original seeding location. These porous membranes were then fixed using paraformaldehyde 11% (Sigma-Aldrich) at room temperature for 30 min and stained using an azure eosin methylene blue 0.4% solution (Giemsa; Sigma-Aldrich) for 3 min.

**Assessment of released bioactive mediators.** The presence of chemotactic and proangiogenic mediators in CM was quantified

using Luminex technology (Millipore, Billerica, USA) following the manufacturer's instructions[32]. The level of 16 mediators, specifically, IL-8, CCL2, MMP9, Chemerin, MIF, CXCL2, CXCL5, CXCL10 (known as chemotactic growth factors) and VEGF-A, VEGF-C, VEGF-D, ANGPT-1, ANGPT-2, ANG, FGF-BASIC, Endostatin (known as angiogenic modulators) was evaluated using the MasterPlex QT 1.0 system (MiraiBio, Alameda, USA) and analyzed using Luminex-100 software version 1.7 (Luminex, Austin, USA).

**Animals**. Ten-week-old female nude immunodeficient mice (NMRI-nu (nu/nu) were obtained from Janvier Labs (Le Genest-Saint-Isle, France). Animal experiments were conducted in accordance with the European Directive 2010/63/EU regarding the protection of animals used for scientific purposes and were approved by the Ethics Committee on Animal Research (APAFIS #14805-2018041119309138 v3).

**Ectopic implantation**. Before each surgical procedure, a dose of buprenorphine (Buprecare®; 0.1 mg/kg animal weight; Axience, Pautin, France) was administrated subcutaneously in each mouse, and the skin was prepared for surgery using an application of povidone-iodine (Betadine®, Vetoquinol, Paris, France). Anesthesia was induced by intraperitoneal administration of ketamine (Ketamine1000®; 100 mg/kg animal weight; Virbac, France) and xylazine (Rompun® 2%; 8 mg/kg animal weight; Bayer Health-Care, Berlin, Germany). Flowing oxygen was delivered using a specific mask for each animal throughout the surgical procedure. Either two (for the assessment of cell proangiogenic potential) or four (for the evaluation of cell viability) symmetrical incisions (each 7.5 mm in length) on both sides of the spine were made on the back of each mouse, and subcutaneous pouches were created. Hydrogels were then carefully and randomly inserted into each pouch. Skin closure was accomplished with an interrupted suture pattern using 4.0 polyglactin 910 sutures (Ethicon, Issy-les-Moulineaux, France). The mice were monitored daily by trained animal-care personnel throughout the postoperative period. Food and water were available *ad libitum* to the animals.

**Assessment of AMG leakage from hydrogels**. A fluorescently labelled AMG was prepared by coupling AMG with Dylight 800 N-hydroxysuccinimide-ester (Thermofischer Scientific) (AMGDL800). Briefly, a Recombinant *Aspergilus Niger* AMG powder (Megazyme) was resuspended in water and the buffer was exchanged against 0.05 M sodium borate pH 8.5 using a Spin-OUT GT600 column (G-biosciences). 4.5 mg AMG in borate buffer was mixed with 250 µg of Dylight 800 NHS incubated at room temperature for 1 h and then dialyzed overnight at 4 °C against 0.1 M sodium citrate buffer pH 5.5. AMGDL800 protein concentration and degree of labelling was determined using a Spectramax ABS+ microplate reader and a spectradrop micro-volume microplate (Molecular Devices). Calculations were done according to the Dylight 800 manufacturer instructions, giving a 0.89 mol dye /mol protein. Hydrogels containing 60 µg of a mix of AMG and AMGDL800 in a ratio of 90/10 were then prepared as aforementioned. Before implantation, the area corresponding to each hydrogel was measured and used to delineate a region of interest beyond which AMG fluorescence was considered to have leaked from the hydrogel. Hydrogels were then implanted ecto-pically in mice. On days 0, 1,2,3,4,7,10, and 14, animals were anesthetized with isofluorane, and fluorescence efficiency in the region of interest was measured non-invasively using a fluorescence imaging system (Ivis, Lumina II, Caliper Life Sciences, Villebon-sur-Yvette, France) according to[34]; These data were normalized to those obtained on day 0 after surgery.

**Assessment of cell viability**. The viability of the Luc-ZS Green-hMSCs contained within the implanted hydrogels was monitored non-invasively using a bioluminescence imaging system on days 1, 7, and 14 (Ivis, Lumina II, Caliper Life Sciences, Villebon-sur-Yvette, France), as previously described[35]. For this purpose, the mice were anesthetized by delivery of 3% isoflurane (Iso-Vet®; Piramal HealthCare, Northumberland, UK) in oxygen, and 100 mg/kg D-luciferin was injected intramuscularly in the area of the hydrogel locations. The mice were then placed in a ventral position inside the detection chamber of the bioluminescence system and maintained under anesthesia. Images were taken every 5 min for 1 h. A region of interest surrounding each hydrogel was manually defined on each image, and the photon flux emitted by each hydrogel was quantified using the *Living Image® 3.2* software (Caliper Life Sciences). The highest BioLu-minescence Intensity (BLI) signal was retained for each mouse and time point tested. The percentage of viable cells post-implantation was determined as the photon fluxes measured at each time point tested normalized with the respective BLI signal acquired the day after surgery (*i.e.*, day-1 of data acquisition). On days 7 and 14, the mice were sacrificed (using an overdose of intracardiac pentobarbital delivery (Dolethal®; Vetoquinol, Paris, France), and the hydrogels were explanted, fixed in 4% paraf-ormaldehyde, and embedded in paraffin. All prepared paraffin sections were processed for human beta-2-microglobulin immu-nodetected (hβ2-MG, a membrane protein that stains human cells) using the Envision+ kit (Dako, Les Ulis, France) and a polyclonal rabbit anti-hβ2-MG (1/1000, Novocastra, Nanterre, France) as the primary antibody as previously described[36].

**Assessment of the proangiogenic potential of hMSCs-containing-hydrogels**. Each 125 µl hydrogel (loaded with $1.25 \times 10^6$ hMSC) was first placed in the center of a silicone tube (Silicone DIA; 4 mm inner diameter, 6 mm outer diameter, 16 mm height; Weber Métaux, France) and then sandwiched between two layers (3 mm height) of fibrin gel added at each tube end. The latter gel, composed of 1 mg/mL fibrin without aprotinin, was completely degraded 24 h after implantation. Blood vessels were visualized by injecting the mice with a radio-opaque polymer compound (Microfil®, Flowtech, Carver, MA, USA) at 14 or 21 days post-implantation. For this purpose, each animal was deeply anesthetized, and the skin from the thorax and the rib cage was incised to access the heart. The left ventricle was catheterized using a 20 G cannula (BD Venflon, Beckson Dick-inson Infusion, Sweden), and the right atrium was cut for blood removal. Each animal was first perfused with isotonic saline (50 mL) containing heparin (100 UI/mL) using a pump (at a 6 mL/min flow rate) for 6 min to rinse the blood from the vas-culature. 14 mL of Microfil® (prepared according to the manu-facturer's instructions using 6.3 mL of Microfil®, 7 mL of the specific diluent, and 0.7 mL of the specific curing agent) were then manually perfused at approximately 2 mL/min to force the Microfil® into the capillary networks without extravasation into the surrounding tissue. The perfused euthanized animals were stored at 4 °C overnight to allow polymerization of the Microfil®; the silicone tubes containing the hydrogels were explanted and fixed in 4% paraformaldehyde overnight. The specimens were then imaged using a Skyscan1172 micro-CT-scanner (Bruker, Evere, Belgium) with voltage 80 kV, current 100 mA, exposure for 85 ms, and 0.3-degree rotation step settings without any filter. The images obtained had 10 µm pixel size. The scanned images were then reconstructed as a stack of slices of each sample using Nrecon software, 16 bits (Bruker, Evere, Belgium). The volumes of interest (VOI; 2 per sample) were set as cylinders overlapping the internal diameter of the silicone tube (4 mm), between the 1st

and the 3rd mm height from the top and the bottom of the silicone tube edges. Indeed, as some hMSC-containing hydrogels tested partly resorbed after implantation, the aforementioned method allowed to define a similar volume of interest for each sample rigorously. New blood vessel volume was reported as the amount of binarized object volume measured within the designated volume of interest within the threshold gray values 120–255 on CTan software (Bruker). Values regarding new blood vessel thickness and numbers were calculated using the abovementioned thresholds. The new blood vessel length was calculated from the formula $V = h*pi*r^2$ on the assumption that the new blood vessel diameter was homogeneous.

*Statistics and reproducibility.* Numerical data were expressed as means + standard of the mean (SEM). Statistical analyses were performed using commercially available software (GraphPad v9.3.1 Software, California Corporation, USA). In vitro experiments were conducted with at least 3 biological replicates per group whereas in vivo experiments were conducted using at least 5 biological replicates for each group of hydrogels tested. The effects of hydrogel group, time of culture/implantation, and their interactions were examined using two-way ANOVAs. For each significant effect, a Tuckey's post-hoc test was conducted. When only two groups of hydrogels were compared at a single time point of analysis (i.e., Supplementary Fig. 1b), a t-test with a Welsh's correction was conducted. For all analyses, a $p$-value < 0,05 was considered significant. For a given group, the significance of the difference between experimental times of culture/implantation are indicated by the α symbol; the significance between groups at the same time of culture/implantation are indicated with $p < 0.05$. $*p < 0.05$; $**p < 0.01$; $***p < 0.001$; and $****p < 0.0001$.

**Reporting summary.** Further information on research design is available in the Nature Portfolio Reporting Summary linked to this article.

### Data availability

Source data behind the graphs can be found in Supplementary Data. All other data are available from the authors on reasonable request.

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

## Acknowledgements

We thank Professor R. Bizios and Doctor D. Logeart-Avramoglou for their valuable comments on the manuscript. We also thank the Technologic Department of Saint-Louis Hospital for confocal images, the SAPC laboratory of the University of Technology of Compiegne for environmental scanning electron microscopy images, the Biology and Biochemistry Department of Lariboisiere Hospital for dosage of glucose, and the CYBIO department of the Cochin Institute for Multiplex analysis. We thank Mr. Stéphane Vasseur and Ms. Maud Chapart, MYOBANK-AFM (Authorization No. AC-2013-1868, Ethics Committee number BB-0033-00012, norma NF S 96-900) for the providing of human muscle biopsies. We would like to acknowledge their funding sources: l'Agence Nationale de la Recherche (ANR-16-ASTR-0012-01 and ANR-12-BSV5-0015-01), the China Scholarship Council (No. 201600160067), and China Postdoctoral Science Foundation (2022M722203). We thank the AniRA-Neurochem technological platform for microelectrode biosensor fabrication. All sketches have been made using Biorender.com.

## Author contributions

D.C. participated in the design and executed in vitro and post-implantation experiments of the present study, collected and analyzed data, and participated in manuscript writing. L.G. participated in the study design, assisted in conducting the in vivo experiments regarding neovascularization and assisted in collecting data and participated in the manuscript writing. P.J., M.A., B.P., L.N., L.-A.D. assisted in designing and collecting data regarding the in vitro experiments about MSC survival. W.P. assisted in in vivo experiments, statistical analysis, and participated in manuscript writing. M.S. and M.A. designed microelectrode biosensors for glucose assessment in the hydrogels tested. DA assisted in the surgery experiments. MH assisted in collecting and analyzing pertinent data from the Boyden chambers experiments. VJT provided myoblasts. BJ and GA designed the prototype of the nutritive hydrogel. PEmmanuel, and L-GV designed and managed the project, provided financial support, and participated in data analysis. PEsther and PH designed and managed the project, provided financial support, participated in data analysis, writing, and editing the manuscript. All authors critically read and approved the present manuscript.

## Competing interests

The authors declare the following competing interests: P.J., P.Esther, L.-G.V., and P.H. disclose that the novel enzyme-controlled, nutritive hydrogels used in the current project, were patented globally (EP14306700; 2014). The other authors declare no competing interests.
