## [Peer Review File · Communications Biology]

Reviewers' comments:

Reviewer #1 (Remarks to the Author):

1. Summary

The manuscript by Cyprien et al. describes a novel hydrogel for delivering and enhancing the survival of human mesenchymal stromal cells (hMSCs) delivered in vivo. The hydrogel, composed of fibrin, starch, and amyloglucosidase (AMG), enhances hMSC survival through providing physiological glucose levels to the hMSC encapsulated in the hydrogel. The authors characterize the gel and demonstrate that soluble glucose concentration is controllable based on then concentration of starch and AMG in the gel. Starch/AMG loaded gels improved hMSC survival in near-anoxic conditions in vitro over glucose-loaded and glucose-free hydrogels, and these results were confirmed in a subcutaneous in vivo mouse model. Additionally, the starch/AMG loaded gels were the most effective at translating the pro-angiogenic effect of hMSC in an in vivo model, where they elicited new blood vessel formation.

2. Impression

This work adequately addresses the well-known problems in the field of hMSC therapies, demonstrating localization of delivery, enhanced viability, and maintenance of a relevant hMSC phenotype (angiogenicity). The hydrogel formulation is novel and well-characterized, and shows impressive increases in both hMSC viability and new angiogenesis in vivo. There are a number of additions that would help to clarify the data presented in this work, outlined in the comments below.

3. Comments

Page 2, Line 11-14. This statement is misleading, as it cites outdated references (ref 10-11, both from 2004) and ignores the large body of literature dedicated to the creation of perfusable and/or vascularized hydrogels (relevant review: Wang et al. *Advances in hydrogel-based vascularized tissues for tissue repair and drug screening*. *Bioactive Materials* 2022)

Page 3, Line 1-2. This statement should be backed by references

Page 3, Line 14-15: Causes for the glucose concentration plateau should be discussed. Has all the starch been hydrolyzed by AMG at 600s? Or is this the time at which new glucose production is matched by diffusion of glucose from the hydrogel? Another experiment to measure remaining starch or glucose diffusion may be necessary to fully describe the lifetime of starch in the hydrogel.

Page 4, Line 13: "increased by 5-fold" implies that the cells are proliferating in the gel, when the data presented suggests that they are surviving but not proliferating in the gel.

Page4, Line 4-8 and Supplemental Fig S1: CD marker expression is a relevant description of hMSC identity and phenotype, however, the authors would make a stronger case for the maintenance of hMSC phenotype function with a demonstration of multipotency (osteogenic, adipogenic, and chondrogenic differentiation) and immunosuppressive potential.

Page 5, Line 14-15: is the 1.5-fold increase in endostatin significantly different in glucose free hydrogels vs starch/AMG hydrogels

Page 7, Line 33-Page 8, Line3: The possibility of disrupted homeostasis and exacerbated immune response should be further explored and relevant literature cited.

Figure 1d: The scale bars in the SEM images are not visible/legible

Figure 5a: This experimental design and schematic are well thought out and nicely visualized. A photograph of the implantation as a supplemental figure would help demonstrate the procedure and the proximity of the hydrogel implant to native blood vessel networks.

Figure 5b: The size of the scale bars in the micro-CT images should be reported in the figure caption

Figure 5b-d: The representative images in b show robust growth of new vessels in both number and length at day 21. However, the new blood vessel numbers in table 5d show averages of less than 0.2 new vessels per mm, with a reported error that extends into the negatives. It could be the case that

the VOI is many mm in diameter and therefore the reported value is not as small as it seems, or that the images in (b) are representative of gels that were vascularized, but other gels weren't. Whatever the case may be, there should be some clarification on how the images in 5b relate to the numbers in 5d.

Reviewer #2 (Remarks to the Author):

The authors proposed a novel way to promote the survival of MSCs encapsulated in a hydrogel by incorporating starch into the hydrogel formulation. They highlight the ischemic conditions of MSC implanted for tissue constructs result in significant dead, but their previous work and others demonstrated that cells exposed to severe, continuous near-anoxia, but not glucose shortage, remained viable and maintained both their in vitro proliferative ability and, most importantly, some of their functions. They suggest fueling the cells via glycolysis will enhance survival under low oxygen concentrations. The main hypothesis was that starch combined with amyloglucosidase (AMG) would provide nutrients/glucose to do so.

Their results demonstrated effectively a hydrogel scaffold with two wt% Starch/AMG improves cell survival in vitro up to 14 days and in vivo for up to 7 days. They also demonstrated the impact of some chemotactic functions. Overall, these results are very promising, and much work continues to be done to improve/optimize this approach.

I would appreciate it if the authors could elaborate and discuss more the enzymatic kinetics, the glucose concentration during the 14 days experiments, and the potential presence and impact of disaccharides from starch hydrolysis.

- 1) Figure 1b. The enzyme reaches a plateau in less than 15 minutes. Why do they reach saturation since there is still a lot of starch in the hydrogel? How did the glucose concentration change during the next hours days during the in vitro experiments?
- 2) Figure 1C. We observed similar saturation with different enzyme concentrations. What is the driver of inhibition in this process?
- 3) What were the criteria for selecting enzyme concentration?
- 4) Figure 1d: Scale bars are not visible/readable in the images. What do the authors hypothesize is the driver for this microporosity? Is this only observed on the surface, or are these caused by a phase separation in the hydrogel or the outer surface?
- 5) For in vitro studies (Figures 2 & 3). Do you have the glucose concentration at Day 7 and Day 14? The initial Glucose 5.5mM control has a higher glucose concentration than the starch groups equilibrium from the kinetics (3-4 mM). The main claim is that the starch acts as a glucose reservoir to fuel the cells. The viability in the Glc 5.5mM control was drastically reduced at Day 7 and Day 14. Could the authors confirm the glucose levels were effectively lower in vitro or that the Start 2%/AMG maintained higher glucose concentration at Day 14?
- 6) Did the authors evaluate the presence of other disaccharides from starch hydrolysis that could alter cell metabolism and impact survival?
- 7) Methods: Please provide more details about how these hydrogels were disintegrated to release the cells.

Reviewer #3 (Remarks to the Author):

ENZYME-CONTROLLED, NUTRITIVE HYDROGEL FOR MESENCHYMAL STROMAL CELL SURVIVAL AND PARACRINE FUNCTIONS, by Denoëud et al

General comments

Mesenchymal stem cells can stand hypoxia but absolutely need energy supply to keep alive. The authors have developed a way to deliver glucose by means of enzymatic lysis of starch. The manuscript by Denoëud et al. is well designed from formulation to in vivo trials.

Viability and paracrine effect are well shown with different assays (chemo-attraction, release factory by luminex technology...)

Discussion: some information is missing about the result difference between glucose production from starch and only glucose. How do you explain this?

Please add the statistics information (significant differences ?, which stat test was done ?) and the number of replicates in figures.

Are you targeting a precise regenerative medicine application (wounds, bones, osteoarthritis, muscle regeneration) ? Because you talk about proangiogenic function which is not applicable for all regenerative medicine?

Comments

p.3 l.14 "glucose production in 400 seconds"... why not only inject glucose then? What is the target release time?

p.3 l.16 "last but not least"

p.3 l.24 Is every condition serum-free?

p.3 l.32 with how much enzymes? How much glucose is available for cells at the end?

p.4. l.9 "more effective than glucose hydrogels.." why? Explain

p.5 l.28 "cessation of glucose production between day 7 and 14..." but if it is produced in 400 seconds?

I don't understand this part

p.5 l.33 what is BLI? Abbreviation

p.6 l.1-5 I don't understand this part. Explain better. A decline in BLI signal is good or bad?

p.7 l.26 How do you explain that after 14 days there are no more cells?

p.8 l. 25 In conclusion, ...

p.14 hydrogel formulation: why all this compounds, maybe explain briefly why you add thrombin, aprotinin etc..

Figures

Fig.1 stat? n=? , justify maybe why you use low glucose medium 1g/L (5.5mM) and not 4.5 g/L of glucose like in the mediums. Where is the scale bar?

Fig.2 stat? n=?, a1) with enzyme and a2) without enzyme how do you explain that they still live without enzyme? Discuss. Can you say number of cells? (y axis)

Fig.5 5d) what are the numbers?

FigS1c) conditions?

Referee expertise:

Referee #1: Defined stem cell culture, biomaterials, scaffolds

Referee #2: Biomaterials for cardiovascular models and therapeutics

Referee #3: nanomedicine, hydrogels

Reviewers' comments:

Reviewer #1 (Remarks to the Author):

1. Summary

The manuscript by Cyprien et al. describes a novel hydrogel for delivering and enhancing the survival of human mesenchymal stromal cells (hMSCs) delivered in vivo. The hydrogel, composed of fibrin, starch, and amyloglucosidase (AMG), enhances hMSC survival through providing physiological glucose levels to the hMSC encapsulated in the hydrogel. The authors characterize the gel and demonstrate that soluble glucose concentration is controllable based on then concentration of starch and AMG in the gel. Starch/AMG loaded gels improved hMSC survival in near-anoxic conditions in vitro over glucose-loaded and glucose-free hydrogels, and these results were confirmed in a subcutaneous in vivo mouse model. Additionally, the starch/AMG loaded gels were the most effective at translating the pro-angiogenic effect of hMSC in an in vivo model, where they elicited new blood vessel formation.

2. Impression

This work adequately addresses the well-known problems in the field of hMSC therapies, demonstrating localization of delivery, enhanced viability, and maintenance of a relevant hMSC phenotype (angioinductivity). The hydrogel formulation is novel and well-characterized, and shows impressive increases in both hMSC viability and new angiogenesis in vivo. There are a number of additions that would help to clarify the data presented in this work, outlined in the comments below.

We would like to thank the Reviewer #1 for his/her careful and thorough reading of this manuscript and for his/her thoughtful comments and constructive suggestions, which helped us revise, and thus, improve the original manuscript. Our responses follow each statement of the Reviewer's comments and are in blue.

3. Comments

Page 2, Line 11-14. This statement is misleading, as it cites outdated references (ref 10-11, both from 2004) and ignores the large body of literature dedicated to the creation of perfusable and/or vascularized hydrogels (relevant review: Wang et al. Advances in hydrogel-based vascularized tissues for tissue repair and drug screening. Bioactive Materials 2022)

We thank reviewer #1 for this helpful comment. We fully agree with the importance of highlighting recent advances in pre-vascularized tissues. We respectfully propose adding the following sentence in the INTRODUCTION section by citing Wang et al. and two additional papers on Page 2 Line 15-16. "Furthermore, pre-vascularized engineering hydrogel has been produced and investigated for the improvement of oxygen and nutrient delivery."

1. Wang Y, Kankala RK, Ou C, Chen A, Yang Z. Advances in hydrogel-based vascularized tissues for tissue repair and drug screening. *Bioact Mater.* 2021;9:198-220. Published 2021 Jul 10. doi:10.1016/j.bioactmat.2021.07.005
2. Sharma D, Sharma A, Hu L, et al. Perfusability and immunogenicity of implantable pre-vascularized tissues recapitulating features of native capillary network. *Bioact Mater.* 2023;30:184-199. Published 2023 Aug 11. doi:10.1016/j.bioactmat.2023.07.023
3. Atlas Y, Gorin C, Novais A, et al. Microvascular maturation by mesenchymal stem cells in vitro improves blood perfusion in implanted tissue constructs. *Biomaterials.* 2021;268:120594. doi:10.1016/j.biomaterials.2020.120594

Page 3, Line 1-2. This statement should be backed by references.

We would like to thank reviewer #1 for drawing our attention to this flaw. We propose to support our statement by citing Schwartzman et al.

1. Schwartzman, C., Zhao, H., Ibarboure, E., Ibrahimova, V., Garanger, E., Lecommandoux, S., Control of Enzyme Reactivity in Response to Osmotic Pressure Modulation Mimicking Dynamic Assembly of Intracellular Organelles. *Adv. Mater.* 2023, 2301856. <https://doi.org/10.1002/adma.202301856>

Page 3, Line 14-15: Causes for the glucose concentration plateau should be discussed. Has all the starch been hydrolyzed by AMG at 600s? Or is this the time at which new glucose production is matched by diffusion of glucose from the hydrogel?

We thank reviewer #1 for this comment as it gives us the opportunity to clarify important aspects of this project. In fact, the hydrogel was placed in 30 ml PBS. After being produced by starch hydrolysis, glucose diffused into the external buffered solution. The plateau in the concentration versus time graph represents the steady state where glucose production equals diffusion outside the hydrogel. We respectfully suggest that the following sentence be added to the DISCUSSION section to explain the glucose production plateau. *“In addition, Glucose production plateaued within 400 to 600 seconds, reflecting the balance between new glucose production and glucose diffusion out of the hydrogel.”* (Page 7 Line 11-13).

Another experiment to measure remaining starch or glucose diffusion may be necessary to fully describe the lifetime of starch in the hydrogel.

We thank reviewer#1 for this comment. Regarding the question of the lifetime of starch in the hydrogel, in vitro tests have shown that the in vitro biodegradation rate of wheat starch is dependent on the concentration of AMG (Fig. 1c). In addition, unpublished studies from our group have shown that a 2% wheat starch/AMG (136 UI/ml) is completely degraded in vitro into glucose within 7 days. However, data on the lifetime of starch in the hydrogel in vivo are limited to qualitative studies in order to comply with the 3R rules and thus limit the number of animals required. To this end, histological sections taken on day 14 post-implantation were stained with lugols iodine and examined microscopically. The data showed that some starch remained after 14 days. In the figure below, residual starch appeared as purplish. This aspect will now be discussed in the Discussion section: *“This decline in performance of starch/AMG hydrogels could be attributed to starch stock exhaustion. However, the presence of starch remnants at day 14 post-implantation (unpublished results) contradicts this hypothesis. This starch/AMG hydrogel performance loss may be attributed to the gradual disappearance of AMG (Fig.4b), which, in turn, leads to a drop in glucose production, an unfulfillment of MSCs energy requirements, and, ultimately to hMSC viability loss. Further research is needed to*

determine whether this gradual disappearance of free AMG is due to its diffusion from the hydrogel or to its degradation by local proteases.” (Page 7 Line 32-33 and Page 8 Line 1-5)

To address the issue of glucose diffusion, we performed some *in vivo* experiments. Glc-free, Glc 5.5 mM, 2% starch and 2% starch/AMG hydrogels were ectopically implanted in mice and the intra-hydrogel glucose concentration was measured at day 7 and 14 after implantation. (Figure 4(a)). The results showed that at both time points, the intra-hydrogel glucose concentration was significantly ($p < 0.0001$) higher in the 2% starch/AMG hydrogels compared to the results obtained with the 2% starch hydrogels (4.6- and 9.4-fold increase, respectively), glucose-free hydrogels (3.9- and 2.9-fold increase, respectively) and 5.5 mM glucose hydrogels (2.3- and 2.7-fold increase, respectively) (Figure 4a). In addition, a significant decrease in the intra-hydrogel glucose concentration within the 2% starch/AMG hydrogels was observed between days 7 and 14, suggesting a cessation of glucose production due to AMG leakage outside the hydrogels.

Page 4, Line 13: “increased by 5-fold” implies that the cells are proliferating in the gel, when the data presented suggests that they are surviving but not proliferating in the gel.

We would like to thank reviewer #1 for drawing our attention to this issue. We agreed and modified this expression to “was 4-fold higher” in the revised manuscript.

Page 4, Line 4-8 and Supplemental Fig S1: CD marker expression is a relevant description of hMSC identity and phenotype, however, the authors would make a stronger case for the maintenance of hMSC phenotype function with a demonstration of multipotency (osteogenic, adipogenic, and chondrogenic differentiation) and immunosuppressive potential.

We thank reviewer#1 for his/her constructive comments. In the present study, the maintenance of the phenotype and functionalities of hMSCs loaded in starch/AMG hydrogels and exposed to near anoxia was demonstrated not only by assessing the expression of CD markers, but also by assessing the proliferative ability of hMSCs. Thus, the proliferative ability of hMSCs loaded in 2% starch/AMG hydrogels after 14 days in near anoxia was similar to that of naive hMSCs cultured under standard culture conditions for 10 days (Fig.S1b), as previously observed in Moya 2018. The maintenance of hMSCs functionalities in the present study is also demonstrated by their superior angiogenic capacity. The maintenance of hMSCs functionalities are also corroborated by previous *in vivo* studies from our group, which demonstrated the maintenance of sheep MSC osteogenic capacity exposed to a hypoxic episode (M. Deschepper 2010). Briefly, sheep MSCs exposed to an ischaemic episode and loaded onto coral particles showed the same bone forming capacity as sheep MSCs cultured under standard culture conditions (i.e. 21% O₂; 10% serum; medium changed every 3 days) after subcutaneous implantation in mice (M. Deschepper 2010). Therefore, although the experiments suggested by reviewer#1 are interesting studies, we feel that they are not necessary to claim a beneficial effect of enzyme-controlled nutritive hydrogel on MSC survival and paracrine functions. For

these reasons, we would respectfully suggest that the reviewer's comment be addressed in the Discussion section (page 7 lines 16-22) by adding the following comment:

Starch/AMG hydrogels significantly ($p < 0.0001$) extended the survival of hMSCs *in vitro* under near anoxia compared to glucose-free and 5.5 mM glucose hydrogels (Fig. 2a). Most importantly, the maintenance of hMSC proliferative ability and CD marker expression after their release from these starch/AMG hydrogels bodes well for their continued *in vivo* therapeutic efficacy (Fig. S1b and c). A demonstration of the maintenance of the multipotential and immunosuppressive character of MSCs under these conditions will enable these results to be extended to all the classically recognised functionalities of MSCs (Page 7, Line 19-22).

Page 5, Line 14-15: is the 1.5-fold increase in endostatin significantly different in glucose free hydrogels vs starch/AMG hydrogels.

We thank reviewer#1 for his/her comments. As mentioned in Figure 3, there is a significant difference in endostatin concentration in 2% starch/AMG hydrogels compared to glucose-free hydrogels (1,5 fold increase; $p < 0,05$). We have added the statement of significance in the revised manuscript.

Page 7, Line 33-Page 8, Line3: The possibility of disrupted homeostasis and exacerbated immune response should be further explored and relevant literature cited.

We would like to thank reviewer#1 for bringing this issue to our attention. We assumed that the foreign body reaction could be an explanation for a disrupted homeostasis and an exacerbated immune response. We proposed to add the following paragraph in the DISCUSSION section, citing relevant literature from Salthouse et al. and Balabiyev et al.

“Another tenable explanation for the observed decrease in hMSC viability over time is that the implanted engineered constructs, which temporarily disrupt homeostasis, induce a local exacerbated immune response whereby the host immune system recognizes hMSCs as foreign, ultimately leading to hMSC death. Although further studies are needed to exclude this hypothesis, the immunocompromised features of the mice, the parrallel disappearance of AMG and MSCs and the key role of glucose in survival, chemotactic and proangiogenic functions of hMSCs make it unlikely. Alternatively, the foreign body reaction may lead to chronic fibrotic capsule formation, preventing hydrogel revascularisation. Again, the improved chemotactic and proangiogenic functions of hMSCs do not support such a scenario.” (Page 8 Line 9-17)

1. Salthouse D, Novakovic K, Hilkens CMU, Ferreira AM. Interplay between biomaterials and the immune system: Challenges and opportunities in regenerative medicine. *Acta Biomater.* 2023;155:1-18. doi:10.1016/j.actbio.2022.11.003
2. Balabiyev A, Podolnikova NP, Kilbourne JA, et al. Fibrin polymer on the surface of biomaterial implants drives the foreign body reaction. *Biomaterials.* 2021;277:121087. doi:10.1016/j.biomaterials.2021.121087

Figure 1d: The scale bars in the SEM images are not visible/legible.

We agree with reviewer#1 on this point. An improvement has been made to improve the visibility. A scale bar in yellow is included, which is 200 μm for the top row and 50 μm for the bottom row.

Figure 5a: This experimental design and schematic are well thought out and nicely visualized. A photograph of the implantation as a supplemental figure would help demonstrate the procedure and the proximity of the hydrogel implant to native blood vessel networks.

We thank the positive feedback on the experimental design and schematic. An additional photograph of the implantation site to further demonstrate the proximity of the hydrogel implant to native blood vessels (Fig 5a).

Figure 5b: The size of the scale bars in the micro-CT images should be reported in the figure caption.

We thank reviewer#1 for this comment. The size of the scale bars (500 μm) is reported in the figure caption.

Figure 5b-d: The representative images in b show robust growth of new vessels in both number and length at day 21. However, the new blood vessel numbers in table 5d show averages of less than 0.2 new vessels per mm, with a reported error that extends into the negatives. It could be the case that the VOI is many mm in diameter and therefore the reported value is not as small as it seems, or that the images in (b) are representative of gels that were vascularized, but other gels weren't. Whatever the case may be, there should be some clarification on how the images in 5b relate to the numbers in 5d.

We are appreciative to reviewer#1 for this helpful comment that help us to clarify this issue. In fact, and surprisingly, the diameters and numbers of the new blood vessels were similar for all groups after implantation in ectopic mouse model. However, the length of the network of new blood vessels formed after 21 days within the starch/AMG hydrogels was more developed compared to the one of glucose-free hydrogels and glucose 5,5 mM hydrogels.

We sincerely hope that our responses adequately address the reviewer's request for clarification, and we are readily available to address any further questions or concerns he/she may have.

Reviewer #2 (Remarks to the Author):

The authors proposed a novel way to promote the survival of MSCs encapsulated in a hydrogel by incorporating starch into the hydrogel formulation. They highlight the ischemic conditions of MSC implanted for tissue constructs result in significant dead, but their previous work and others demonstrated that cells exposed to severe, continuous near-anoxia, but not glucose shortage, remained viable and maintained both their in vitro proliferative ability and, most importantly, some of their functions. They suggest fueling the cells via glycolysis will enhance survival under low oxygen concentrations. The main hypothesis was that starch combined with amyloglucosidase (AMG) would provide nutrients/glucose to do so.

Their results demonstrated effectively a hydrogel scaffold with two wt% Starch/AMG improves cell survival in vitro up to 14 days and in vivo for up to 7 days. They also demonstrated the impact of some chemotactic functions. Overall, these results are very promising, and much work continues to be done to improve/optimize this approach.

We would like to thank the Reviewer #2 for his/her careful and thorough reading of this manuscript and for his/her thoughtful comments and constructive suggestions, which helped us revise, and thus, improve the original manuscript. Our responses follow each statement of the Reviewer's comments and are in blue.

I would appreciate it if the authors could elaborate and discuss more the enzymatic kinetics, the glucose concentration during the 14 days experiments, and the potential presence and impact of disaccharides from starch hydrolysis.

We thank reviewer#2 for this constructive comment.

In the present study, we observed that AMG, at the origin of glucose production in the starch2%/AMG hydrogels, progressively disappeared within 7 to 10 days post-implantation resulting in glucose concentration of 4.78 ± 0.28 and 3.68 ± 0.41 at day 7 and 14.

In 5.5mM Glc hydrogels, free glucose, is highly diffusive. For instance, it is released from 5.5mM hydrogels within 4 hours in vitro.

It is our belief that this high diffusivity is at the origin of the modest effect of 5.5mM Glc hydrogels. In starch/AMG hydrogels,

The response regarding the potential presence and impact of disaccharides from starch hydrolysis is given in point 6.

1) Figure 1b. The enzyme reaches a plateau in less than 15 minutes. Why do they reach saturation since there is still a lot of starch in the hydrogel? How did the glucose concentration change during the next hours days during the in vitro experiments?

We thank reviewer#2 for this comment as it gives us the opportunity to clarify important aspects of this project. We will respond separately to these two questions for the sake of clarity.

Question 1 “Why do they reach saturation since there is still a lot of starch in the hydrogel?”

In fact, tested hydrogels were placed in 30 ml PBS. In these conditions, after being produced by starch hydrolysis, glucose diffused into the external buffered solution. The plateau in the concentration versus time graph represents the steady state where glucose production equals diffusion outside the hydrogel. We respectfully suggest that the following sentence be added

to the DISCUSSION section to explain the glucose production plateau. “In addition, Glucose production plateaued within 400 to 600 seconds, reflecting the balance between new glucose production and glucose diffusion out of the hydrogel.” (Page 7 Line 11-13).

Question 2: “How did the glucose concentration change during the next hours days during the in vitro experiments?”

The question is very similar to point 5 of reviewer#2 “For in vitro studies (Figures 2 & 3). Do you have the glucose concentration on day 7 and day 14?” For the sake of brevity, we refer reviewer #2 to our answer in point 5.

2) Figure 1C. We observed similar saturation with different enzyme concentrations. What is the driver of inhibition in this process?

We thank reviewer#2 for this comment. As explained in the answer to question, in our experimental model, after being produced by starch hydrolysis, glucose diffused into the external buffered solution. The plateau in the concentration versus time graph represents the steady state where glucose production equals diffusion outside the hydrogel.

3) What were the criteria for selecting enzyme concentration?

We thank reviewer #2 for this critical comment. The enzyme concentration is selected based on the patent “Time-controlled glucose releasing hydrogels and applications” (CNRS, EP 14306700, US-9931433-B2)” and preliminary experiments. However, it is fair to recognize that at this stage, it is not yet optimum. This limitation was underscored in the discussion section by stating “However, this strategy is still in its infancy and to reach their full potential, hydrogels will have to be fully resorbable and tailored to the dose of MSCs and the duration of treatment required, as with any pharmacological treatment.”. If reviewer #2 feels this is not sufficient, we will be happy to further comment this aspect.

4) Figure 1d: Scale bars are not visible/readable in the images. What do the authors hypothesize is the driver for this microporosity? Is this only observed on the surface, or are these caused by a phase separation in the hydrogel or the outer surface?

We would like to thank reviewer#2 for spotting the quality of images and the very helpful comment. New images with readable scale bars have been provided in the revised manuscript. Our possible explanation for the observed microporosity is that starch and the released glucose may probably act as a pore-forming agent (Nikolopoulos N et al. 2023 and Smith BT et al. 2018). On Fig 1d, only the surface of the hydrogels was observed using SEM. However, when we focus into external pores, we can see same structure (microporosity) inside the hydrogels.

1. Nikolopoulos N, Parker LA, Wickramasinghe A, van Veenhuizen O, Whiting G, Weckhuysen BM. Addition of Pore-Forming Agents and Their Effect on the Pore Architecture and Catalytic Behavior of Shaped Zeolite-Based Catalyst Bodies. *Chem Biomed Eng.* 2023;1(1):40-48. Published 2023 Mar 6. doi:10.1021/cbmi.2c00009
2. Smith BT, Lu A, Watson E, et al. Incorporation of fast dissolving glucose porogens and poly(lactic-co-glycolic acid) microparticles within calcium phosphate cements for bone tissue regeneration. *Acta Biomater.* 2018;78:341-350. doi:10.1016/j.actbio.2018.07.054

5) For in vitro studies (Figures 2 & 3). Do you have the glucose concentration at Day 7 and Day 14? The initial Glucose 5.5mM control has a higher glucose concentration than the starch groups equilibrium from the kinetics (3-4 mM). The main claim is that the starch acts as a glucose reservoir to fuel the cells. The viability in the Glc 5.5mM control was drastically

reduced at Day 7 and Day 14. Could the authors confirm the glucose levels were effectively lower in vitro or that the Starch 2%/AMG maintained higher glucose concentration at Day 14?

We are unsure of the significance of this question and we will do our best to respond to reviewer#2. The glucose concentration measured in the hydrogel supernatant in vitro was $0,48 \pm 0,06\text{mM}$ and $0,30 \pm 0,01\text{mM}$ for Glc 5.5mM hydrogels and $5,6 \pm 0,48\text{mM}$ and $7,73 \pm 0,53\text{mM}$ for starch2%/AMG hydrogels on day 7 and 14. It is reasonable to assume that these glucose concentrations measured in the supernatants reflect the intra-hydrogel glucose concentration as these experimental systems are closed systems and glucose a highly diffusible molecule. In fact, inhouse unpublished experiments revealed that 100% glucose, is released within 4 hours from 5.5mM Glc hydrogels in vitro. Based on this assumption, it is reasonable to propose that the glucose levels to which the MSCs were exposed were effectively lower in the Glc 5.5mM hydrogels than in the starch 2%/AMG. These differences in glucose levels resulted in enhanced survival, chemotactic and proangiogenic functions of hMSCs exposed to near-anoxia in starch 2%/AMG hydrogels compared to Glc 5.5mM hydrogels. We hope we have answered the reviewer's question and remain at his/her disposal for any further clarification.

6) Did the authors evaluate the presence of other disaccharides from starch hydrolysis that could alter cell metabolism and impact survival?

We thank reviewer#2 for this constructive comment. In the present study, we focus on the hydrolysis of glucose acting as a direct energy substrate for MSCs. Unfortunately, we did not look for the presence of other disaccharides. However, such disaccharides are not expected as the overall result of AMG activity is the complete breakdown of starch into its constituent glucose molecules. This is best demonstrated by unpublished in vitro studies from our group showing that hydrogels containing 2% wheat starch and 136 UI/ml AMG are completely degraded into glucose within 7 days. Nevertheless, we cannot exclude the possibility that other disaccharides may temporarily appear as intermediates. This aspect will be investigated in future studies.

7) Methods: Please provide more details about how these hydrogels were disintegrated to release the cells.

We thank reviewer#2 for his interest in the method details. The details are provided in Supplementary Information S3.3 as follows "Hydrogels were then digested, and hMSCs were detached from hydrogels using trypsin-EDTA for 20 min. Then PBS containing 2% bovine serum albumin (BSA, Sigma-Aldrich) was then added to stop the chemical action of trypsin. After centrifugation (at 3,500xg for 5 min), the hMSCs were re-suspended in fresh PBS".

We sincerely hope that our responses adequately address the reviewer's request for clarification, and we are readily available to address any further questions or concerns he/she may have.

Reviewer #3 (Remarks to the Author):

ENZYME-CONTROLLED, NUTRITIVE HYDROGEL FOR MESENCHYMAL STROMAL CELL SURVIVAL AND PARACRINE FUNCTIONS, by Denoed et al

General comments

Mesenchymal stem cells can stand hypoxia but absolutely need energy supply to keep alive. The authors have developed a way to deliver glucose by means of enzymatic lysis of starch. The manuscript by Denoed et al. is well designed from formulation to in vivo trials. Viability and paracrine effect are well shown with different assays (chemo-attraction, release factory by luminex technology...)

We would like to thank the Reviewer #1 for his/her careful and thorough reading of this manuscript and for his/her thoughtful comments and constructive suggestions, which helped us revise, and thus, improve the original manuscript. Our responses follow each statement of the Reviewer's comments and are in blue.

Discussion: some information is missing about the result difference between glucose production from starch and only glucose. How do you explain this?

We thank reviewer#3 for his/her constructive comment. Glucose is a highly diffusive molecule. Indeed, unpublished in-house experiments have shown that 100% glucose is released from 5.5mM Glc hydrogels in vitro within 4 hours. It is very reasonable to assume that glucose released from 5.5mM Glc hydrogels or produced by 2% starch/AMG hydrogels will have the same rate of diffusion out of the hydrogels. However, the intra-hydrogel glucose concentration of the starch/AMG hydrogels was significantly ($p < 0.0001$) higher than that of the 5.5 mM glucose hydrogels (2.3 and 2.7 fold increase, respectively) (Fig. 4a), strongly suggesting that the difference in results between glucose production from starch and from glucose alone is related to the lifetime of starch and AMG in the hydrogels. We will briefly summarise what is known about these two parameters.

- Concerning the lifetime of starch, we have performed in vitro tests which showed that the in vitro biodegradation rate of wheat starch depends on the concentration of AMG (Fig. 1c). In addition, unpublished studies from our group have shown that a 2% wheat starch/AMG (136 UI/ml) is completely degraded to glucose in vitro within 7 days. However, data on the lifetime of starch in the hydrogel in vivo is limited to a qualitative study in order to comply with the 3Rs and therefore limit the number of animals required. To this end, histological sections taken on day 14 post-implantation were stained with Lugol and examined microscopically. The data showed that some starch remained after 14 days. In the figure below, residual starch appears as purple).
- Regarding the lifetime of AMG, monitoring the fate of fluorescently labelled AMG showed its quasi-disappearance from the hydrogels at day 7 of implantation (Fig. 4b), which translates into a significant decrease in the intra-hydrogel glucose concentration within the 2% starch/AMG hydrogels between day 7 and 14, suggesting a cessation of glucose production.

This aspect is now discussed in the discussion section: *“This decline in performance of starch/AMG hydrogels could be attributed to starch stock exhaustion. However, the presence of starch remnants at day 14 post-implantation (unpublished results) contradicts this hypothesis. This starch/AMG hydrogel performance loss may be attributed to the gradual disappearance of AMG (Fig.4b), which, in turn, leads to a drop in glucose production, an unfulfillment of MSCs energy requirements, and, ultimately to hMSC viability loss. Further*

research is needed to determine whether this gradual disappearance of free AMG is due to its diffusion from the hydrogel or to its degradation by local proteases.” (Page 7 Line 32-33 and Page 8 Line 1-5) We sincerely hope that our response adequately addresses the reviewer's request for clarification, and we are readily available to address any further questions or concerns they may have.

Please add the statistics information (significant differences ?, which stat test was done ?) and the number of replicates in figures.

We are grateful to Reviewer #3 for pointing out the statistics information issue. A general information of statistics analyses is provided in the METHODS section. The number of replicates is given in each figure legend in the revised manuscript.

Are you targeting a precise regenerative medicine application (wounds, bones, osteoarthritis, muscle regeneration) ? Because you talk about proangiogenic function which is not applicable for all regenerative medicine?

We would like to thank reviewer #3 for his comment regarding the precise regenerative medicine application. We fully agreed that a pro-angiogenic function is not applicable for all regenerative medicine. The bone regeneration is our preferred application based on our previous investigations. Additionally, the regenerative medicine concerning the importance of vascularization is our potential directions, such as wound healing.

Comments

p.3 l.14 “glucose production in 400 seconds”... why not only inject glucose then? What is the target release time?

The aim of this measurement is to validate our starch/AMG system is effective in producing glucose. The ultimate target release time is to provide glucose release until the host vascularization is reached. Only injecting glucose can provide glucose for short-term, however, a long-lasting glucose delivery requires a continuously glucose release from the starch/AMG system.

p.3 l16 “last but not least”

We thank reviewer #3 for his/her comment and replace “last but not least” by “Moreover”.

p.3 l.24 Is every condition serum-free?

Yes, all conditions involving the assessment of MSC survival and functionalities are serum-free in present study.

p.3 l.32 with how much enzymes? How much glucose is available for cells at the end?

The enzyme concentration loaded in starch2%/AMG hydrogels is 1.95 mg/ml. The glucose concentration within cell-free Glc free, Glc5.5mM, Starch 2% and Starch 2%/AMG hydrogels after 7 and 14 days subcutaneous implantation have been investigated and are detailed in Fig 4a.

p.4. 19 “more effective than glucose hydrogels..” why? Explain

We thank reviewer #3 for this comment as it gives us the opportunity to clarify important aspects of this project. To address this issue, we measured the glucose concentration in the hydrogel supernatant in vitro based on the assumption that the glucose concentration in the hydrogel supernatant reflects the intra-hydrogel glucose concentration for two reasons: (i) our experimental system is a closed system; and (ii) glucose is a highly diffusible molecule. The results showed that the hydrogel supernatant glucose concentration was $0.48 \pm 0.06\text{mM}$ and

0.30 ± 0.01mM for Glc 5.5mM hydrogels and 5.6 ± 0.48mM and 7.73 ± 0.53mM for starch2%/AMG hydrogels on days 7 and 14. These data strongly suggest that the glucose levels to which the MSCs were exposed were effectively lower in the Glc 5.5mM hydrogels than in the starch 2%/AMG hydrogels. These differences in glucose levels to which hMSCs are exposed resulted in enhanced survival, chemotactic and proangiogenic functions of hMSCs exposed to near-anoxia in starch 2%/AMG hydrogels compared to Glc 5.5mM hydrogels. We hope we have answered the reviewer's question and remain at his disposal for any further clarification.

p.5 l.28 “cessation of glucose production between day 7 and 14...” but if it is produced in 400 seconds? I don’t understand this part

We thank reviewer #3 for his/her comment as it gives us the opportunity to clarify important aspects of this project. We will explain separately the reasons for this apparent contradictory observations and would like to underscore that the experiment pertaining to “cessation of glucose production between day 7 and 14...” was carry out *in vivo* while the experiment about ...” but if it is produced in 400 seconds? was carried out *in vitro*.

- Meaning and interpretation of “cessation of glucose production between day 7 and 14...”: Glucose production is attributed to AMG activity in 2% starch/AMG hydrogels. The observed cessation of glucose production *in vivo* between days 7 and 14 is the result of the disappearance of AMG from 2% starch/AMG hydrogels implanted subcutaneously in mice. This was demonstrated by monitoring the fate of fluorescence-labelled AMG loaded into acellular hydrogels, which showed evidence of quasi-disappearance at day 7 of implantation (Figure 4b).
- Meaning and interpretation of “...” but if it is produced in 400 seconds? “: In our *in vitro* system, tested hydrogels were placed in 30 ml PBS. In these conditions, after being produced by starch hydrolysis, glucose diffused into the external buffered solution. The plateau in the concentration versus time graph represents the steady state where glucose production equals diffusion outside the hydrogel. We respectfully suggest that the following sentence be added to the DISCUSSION section to explain the glucose production plateau. “In addition, Glucose production plateaued within 400 to 600 seconds, reflecting the balance between new glucose production and glucose diffusion out of the hydrogel.” (Page x Line y).

p.5 l.33 what is BLI? Abbreviation

We thank reviewer #3 for his/her comment. The BLI is the abbreviation for BioLuminescence Intensity. The meaning of this abbreviation can now be found Page 19 Line 11.

p.6 l.1-5 I don’t understand this part. Explain better. A decline in BLI signal is good or bad?

We thank reviewer #3 for his/her comment. The bioluminescence signal represent live cell activity, thus, a decline in BLI signal implies less viable cell.

p.7 l.26 How do you explain that after 14 days there are no more cells?

We thank reviewer #3 for his/her comment. A decrease in viable cells within the hydrogel after implantation for 14 days is one of the limitations of the present study. An observed leakage of AMG probably led to a decrease in glucose, resulting in cell death. Therefore, optimising the current starch/AMG system to improve cell survival for long term (more than 14 days) is our future direction.

p.8 l. 25 In conclusion, ...

We are unsure of the significance of this comment but will be happy to modify at the reviewer's request.

p.14 hydrogel formulation: why all this compounds, maybe explain briefly why you add thrombin, aprotinin etc..

We would like to thank reviewer#3 for his interest in the hydrogel formulation. Details about the functions of each component is provided as follows:

- Fibrinogen: a protein made up of fibrinopeptides, playing an essential role in the maintenance of homeostasis and blood clot formation (Kostelansky et al., 2004). This protein is used here to be associated with thrombin.
- Thrombin: an enzyme that specifically hydrolyzes fibrinopeptides of fibrinogen. Thrombin is used here to synthesize the three-dimensional fibrin network, also known as fibrin hydrogel (Hantgan and Hermans, 1979). The fibrin network promotes adhesion and proliferation of many cell types (Zhao et al., 2008; Janmey et al., 2009), including MSCs (Bensaïd et al., 2003).
- Aprotinin: a protease inhibitor with anti-fibrinolytic properties. It is used here to maintain the fibrin hydrogel over time, notably by limiting its degradation by proteases (Jockhoveel et al., 2001).
- Wheat starch: starch is the storage form of glucose in plants, composed of two types of glucose polymers linked by $\alpha(1-4)$ glucoside bonds: amylose (30%) and amylopectin (70%). Amylose is made up of a linear chain of glucose (Buléon et al., 1998), while amylopectin is made up of branched glucose chains linked by $\alpha(1-6)$ bonds to the main linear chain (Perez et al., 2010).
- Amyloglucosidase: a finishing enzyme that successively hydrolyzes starch's $\alpha(1-4)$ glucoside linkages from non-reducing ends, as well as $\alpha(1-6)$ linkages, resulting in the disconnection of amylopectin in starch (Pazur et al., 1959).

Figures

Fig.1 stat? n=? , justify maybe why you use low glucose medium 1g/L (5.5mM) and not 4.5 g/L of glucose like in the mediums. Where is the scale bar?

Data were analyzed statistically using analysis of variance and the Bonferroni post-hoc test and the Mann-Whitney test for parametric and non-parametric tests, respectively. For all analyses, the confidence interval was set at 95% and the significant level at $p < 0.05$. The number of replicates and scale bar are included in the revised manuscript. We chose low glucose medium 1 g/L (5.5mM) because this concentration is a physiological relevant glucose concentration, the high glucose concentration 4.5 g/L represents a hyperglycemic condition.

Fig.2 stat? n=?, a1) with enzyme and a2) without enzyme how do you explain that they still live without enzyme? Discuss. Can you say number of cells? (y axis)

The statistics information and number of replicates are included in the revised manuscript. A detectable glucose production from hydrogel without enzyme could be an explanation for the observed viable cells. Additionally, the glucose release from hydrogel without enzyme is because a spontaneous hydrolysis within the hydrogel. We can say "number of cells" on the y axis.

Fig.5 5d) what are the numbers?

The number of samples is 8 in fig.5d.

FigS1c) conditions?

The near-anoxia conditions represent a 0.1% oxygen, glucose-free, and serum-free. The standard conditions represent the standard cell culture conditions, including 21% oxygen, 1 g/L glucose, and 10% serum.

We sincerely hope that our responses adequately address the reviewer's request for clarification, and we are readily available to address any further questions or concerns he/she may have.

REVIEWERS' COMMENTS:

Reviewer #1 (Remarks to the Author):

1. Summary

The manuscript by Cyprien et al. describes a novel hydrogel for delivering and enhancing the survival of human mesenchymal stromal cells (hMSCs) delivered in vivo. The hydrogel, composed of fibrin, starch, and amyloglucosidase (AMG), enhances hMSC survival through providing physiological glucose levels to the hMSC encapsulated in the hydrogel. The authors characterize the gel and demonstrate that soluble glucose concentration is controllable based on then concentration of starch and AMG in the gel. Starch/AMG loaded gels improved hMSC survival in near-anoxic conditions in vitro over glucose-loaded and glucose-free hydrogels, and these results were confirmed in a subcutaneous in vivo mouse model. Additionally, the starch/AMG loaded gels were the most effective at translating the pro-angiogenic effect of hMSC in an in vivo model, where they elicited new blood vessel formation.

2. Impression

This work adequately addresses the well-known problems in the field of hMSC therapies, demonstrating localization of delivery, enhanced viability, and maintenance of a relevant hMSC phenotype (angiogenicity). The hydrogel formulation is novel and well-characterized, and shows impressive increases in both hMSC viability and new angiogenesis in vivo.

The Authors have adequately addressed the comments for the original manuscript, and the changes made to the text are sufficient. There are a few outstanding points that should be addressed:

1. Regarding the lifetime of starch from the hydrogel, it would be preferable to include the presence of starch remnants as a supplemental figure rather than unpublished results. Otherwise, the discussion of the potential dynamics governing starch production/concentration is sufficient and well-discussed in the added text. The glucose concentration experiments also add clarity to this section.

2. Regarding hMSC function and phenotype retention: The authors' rebuttal makes strong points for the type of characterization done in this study, namely, that proliferative ability and angiogenicity are relevant markers for the model system being investigated. The added sentence to the discussion section makes this clear. While this may be included in a follow-up study, adding these classically recognized characterizations would strengthen this manuscript and increase the breadth of in vivo systems in which this hydrogel may be applicable.

3. Regarding Figure 5b-d: The original review stated:

"Figure 5b-d: The representative images in b show robust growth of new vessels in both number and length at day 21. However, the new blood vessel numbers in table 5d show averages of less than 0.2 new vessels per mm, with a reported error that extends into the negatives. It could be the case that the VOI is many mm in diameter and therefore the reported value is not as small as it seems, or that the images in (b) are representative of gels that were vascularized, but other gels weren't. Whatever the case may be, there should be some clarification on how the images in 5b relate to the numbers in 5d."

The authors' rebuttal states that the length of the network of new blood vessels after 21 days is higher than in the other conditions, resulting in the differences seen in the representative images. This information should be reported in Figure 5d, and added to the discussion of this figure to make it clear in the manuscript what is causing the difference in the representative images from each condition in 5b and the quantification of blood vessel characteristics in 5d.

Reviewer #2 (Remarks to the Author):

Thanks for addressing the questions and comments from all reviewers. My only further recommendation would be to include a brief rationale for the selected enzyme concentration in the manuscript. This is not really addressed in the discussion and would provide the reader with a clear idea of the rationale even though this is still in its early stages.

Reviewer #3 (Remarks to the Author):

The authors have taken into account the reviewers' comments. They have answered correctly and modified the manuscript accordingly

REVIEWERS' COMMENTS:

Reviewer #1 (Remarks to the Author):

1. Summary

The manuscript by Cyprien et al. describes a novel hydrogel for delivering and enhancing the survival of human mesenchymal stromal cells (hMSCs) delivered in vivo. The hydrogel, composed of fibrin, starch, and amyloglucosidase (AMG), enhances hMSC survival through providing physiological glucose levels to the hMSC encapsulated in the hydrogel. The authors characterize the gel and demonstrate that soluble glucose concentration is controllable based on then concentration of starch and AMG in the gel. Starch/AMG loaded gels improved hMSC survival in near-anoxic conditions in vitro over glucose-loaded and glucose-free hydrogels, and these results were confirmed in a subcutaneous in vivo mouse model. Additionally, the starch/AMG loaded gels were the most effective at translating the pro-angiogenic effect of hMSC in an in vivo model, where they elicited new blood vessel formation.

2. Impression

This work adequately addresses the well-known problems in the field of hMSC therapies, demonstrating localization of delivery, enhanced viability, and maintenance of a relevant hMSC phenotype (angiogenicity). The hydrogel formulation is novel and well-characterized, and shows impressive increases in both hMSC viability and new angiogenesis in vivo.

The Authors have adequately addressed the comments for the original manuscript, and the changes made to the text are sufficient. There are a few outstanding points that should be addressed:

We would like to thank the Reviewer #1 for his/her careful and thorough reading of this manuscript and for his/her thoughtful comments and constructive suggestions, which helped us revise, and thus, improve the original manuscript. Our responses follow each statement of the Reviewer's comments and are in blue.

1. Regarding the lifetime of starch from the hydrogel, it would be preferable to include the presence of starch remnants as a supplemental figure rather than unpublished results. Otherwise, the discussion of the potential dynamics governing starch production/concentration is sufficient and well-discussed in the added text. The glucose concentration experiments also add clarity to this section.

We thank reviewer #1 for this constructive comment and have include the the presence of starch remnants as Supplemental Fig. 3 in the revised manuscript.

2. Regarding hMSC function and phenotype retention: The authors' rebuttal makes strong points for the type of characterization done in this study, namely, that proliferative ability and angiogenicity are relevant markers for the model system being investigated. The added sentence to the discussion section makes this clear. While this may be included in a follow-up study, adding these classically recognized characterizations would strengthen this manuscript and increase the breadth of in vivo systems in which this hydrogel may be applicable.

We sincerely appreciate the insights provided by reviewer #1 and their acknowledgement of the value of the characterisation performed in this study. In particular, we are grateful for their recognition that proliferative capacity and angiogenicity are relevant markers for the model system under investigation. While we agree that demonstrating the preservation of the

multipotential and immunosuppressive properties of MSCs under these conditions would allow the extension of these findings to all traditionally recognised functions of MSCs, we respectfully assert that these additional experiments may not be essential to demonstrate the beneficial effects of the enzyme-controlled nutrient hydrogel on MSC survival and paracrine functions. In light of these considerations, we have thoughtfully addressed the reviewer's comment in the Discussion section.

“Starch/AMG hydrogels significantly ($p < 0.0001$) extended the survival of hMSCs in vitro under near anoxia compared to glucose-free and 5.5 mM glucose hydrogels (Fig. 2a). Most importantly, the maintenance of hMSC proliferative ability and CD marker expression after their release from these starch/AMG hydrogels bodes well for their continued in vivo therapeutic efficacy (Supplementary Fig. 1b and c). A demonstration of the maintenance of the multipotential and immunosuppressive character of MSCs under these conditions will enable these results to be extended to all the classically recognised functionalities of MSCs.” (Page 7, Line 16-22).

3. Regarding Figure 5b-d: The original review stated:

“Figure 5b-d: The representative images in b show robust growth of new vessels in both number and length at day 21. However, the new blood vessel numbers in table 5d show averages of less than 0.2 new vessels per mm, with a reported error that extends into the negatives. It could be the case that the VOI is many mm in diameter and therefore the reported value is not as small as it seems, or that the images in (b) are representative of gels that were vascularized, but other gels weren't. Whatever the case may be, there should be some clarification on how the images in 5b relate to the numbers in 5d.”

The authors' rebuttal states that the length of the network of new blood vessels after 21 days is higher than in the other conditions, resulting in the differences seen in the representative images. This information should be reported in Figure 5d, and added to the discussion of this figure to make it clear in the manuscript what is causing the difference in the representative images from each condition in 5b and the quantification of blood vessel characteristics in 5d.

We thank reviewer #1 for his/her constructive comments. The average new blood vessel length are shown below and in the table from Fig.5d. Results showed that the new blood vessels formed near the 2% starch/AMG hydrogels were significantly longer than those found near the glucose-free hydrogels on day 14 ($p < 0.05$) and 21 ($p < 0.01$) and the 5.5 mM glucose hydrogels on day 21 ($p < 0.01$), respectively (Fig. 5b, 5d).

Neovessel Length (μm)			
Type of hydrogels	Glc Free	Glc 5.5mM	Starch2% / AMG
DAY 14	51 \pm 13	57 \pm 12	121 \pm 24*
DAY 21	43 \pm 11	63 \pm 7	154 \pm 39**

As requested by the reviewer, we add the following sentences in the appropriate sections:

- *Result section entitled “The starch/AMG hydrogels are more effective than glucose hydrogels in increasing the proangiogenic functions of hMSCs after subcutaneous implantation”*

“Similarly, the new blood vessels formed near the 2% starch/AMG hydrogels were significantly longer than those found near the glucose-free hydrogels on day 14 ($p < 0.05$) and 21 ($p < 0.01$) and the 5.5 mM glucose hydrogels on day 21 ($p < 0.01$), respectively (Fig. 5b, 5d).” (Page 6 line 23-26)

- *Discussion section*

“Interestingly, this increase in vessel volume results in an increase in vessel length rather than an enlargement in vessel diameter.” (Page 8 line 27-28)

- *Materials and Methods section “Assessment of the proangiogenic potential of hMSCs-containing-hydrogels”*

“The new blood vessel length was calculated from the formula $V=h\pi*r^2$ on the assumption that the new blood vessel diameter was homogeneous.”* (Page 17 line 18-19)

- *Legends from Figure 5. The following highlighted phrase.*

(d). Table summarizing the number, diameter **and length of** newly formed blood vessel into the VOIs of either glucose-free, 5.5 mM glucose or starch 2%/AMG hydrogels after subcutaneous implantation in nude mice (n=8).

We sincerely hope that our responses adequately address the reviewer's request for clarification, and we are readily available to address any further questions or concerns he/she may have.

Reviewer #2 (Remarks to the Author):

Thanks for addressing the questions and comments from all reviewers. My only further recommendation would be to include a brief rationale for the selected enzyme concentration in the manuscript. This is not really addressed in the discussion and would provide the reader with a clear idea of the rationale even though this is still in its early stages.

We would like to thank the Reviewer #2 for his/her careful and thorough reading of this manuscript and for his/her thoughtful comments and constructive suggestions, which helped us revise, and thus, improve the original manuscript. Our responses follow each statement of the Reviewer's comments and are in blue.

In the present study, glucose production inside 2% starch/AMG hydrogels was linearly proportional to AMG concentration (Fig. 1c). Among the concentrations tested, a 0.1% AMG concentration was chosen because it produced a slower degradation of starch *in vitro* (see figure below). Along the same lines, a progressive degradation of starch was observed using similar AMG concentrations in a HEPES buffer (J. Boisselier thesis). As requested by the reviewer, we add the following sentence in Materials and Methods section "Preparation of hydrogels":

"A concentration of 0.1% AMG was chosen because, of the AMG concentrations tested ranging from 136 U/ml to 1092 U/ml, a concentration of 136 U/ml AMG resulted in the slower degradation of starch in vitro." (Page 11 line 22-24)

Ref : J. Boisselier « Mise en œuvre d'un système de confinement et de délivrance moléculaire pour la production in situ de glucose au sein d'un hydrogel conçu pour l'ingénierie tissulaire defended at cergy-Pontoise 11/09/2016.

We sincerely hope that our responses adequately address the reviewer's request for clarification, and we are readily available to address any further questions or concerns he/she may have.

Reviewer #3 (Remarks to the Author):

The authors have taken into account the reviewers' comments. They have answered correctly and modified the manuscript accordingly

We would like to thank reviewer #3 again for his/her constructive comments and appreciate his/her satisfaction with our responses.